# Coordination between ESCRT function and Rab conversion during endosome maturation

Daniel P Ott[1], Samit Desai[1], Jachen A Solinger [iD][1], Andres Kaech[2] & Anne Spang [iD][1]✉

## Abstract

The endosomal pathway is essential for regulating cell signaling and cellular homeostasis. Rab5 positive early endosomes receive proteins from the plasma membrane. Dependent on a ubiquitin mark on the protein, they will be either recycled or sorted into intraluminal vesicles (ILVs) by endosomal sorting complex required for transport (ESCRT) proteins. During endosome maturation Rab5 is replaced by Rab7 on endosomes that are able to fuse with lysosomes to form endolysosomes. However, whether ESCRT-driven ILV formation and Rab5-to-Rab7 conversion are coordinated remains unknown. Here we show that loss of early ESCRTs led to enlarged Rab5 positive endosomes and prohibited Rab conversion. Reduction of ubiquitinated cargo alleviated this phenotype. Moreover, ubiquitinated proteins on the endosomal limiting membrane prevented the displacement of the Rab5 guanine nucleotide exchange factor (GEF) RABX-5 by the GEF for Rab7, SAND-1/CCZ-1. Overexpression of Rab7 could partially overcome this block, even in the absence of SAND-1 or CCZ1, suggesting the presence of a second Rab7 GEF. Our data reveal a hierarchy of events in which cargo corralling by ESCRTs is upstream of Rab conversion, suggesting that ESCRT-0 and ubiquitinated cargo could act as timers that determine the onset of Rab conversion.

**Keywords** Endosomes; Live Cell Imaging; Mammalian Cells; Rab GTPases; *C. elegans*
**Subject Category** Membranes & Trafficking

## Introduction

Cells communicate via exo- and endocytosis with their environment. These processes are also essential for the regulation of nutrient uptake and membrane domain formation at the plasma membrane (Dixson et al, 2023; Palm and Thompson, 2017; Rahmani et al, 2019; Sigismund et al, 2012). Once material has been taken up from the plasma membrane through the formation of endocytic vesicles, these vesicles can undergo kiss-and-run on sorting endosomes for fast recycling of proteins to the plasma membrane and eventually fuse with early endosomes (Solinger et al, 2020; Solinger et al, 2022; Solinger and Spang, 2022). The early endosome will become a sorting/maturing endosome to recycle proteins to the plasma membrane and the Golgi apparatus, and to sort proteins into intraluminal vesicles (ILVs) for degradation in endolysosomes/lysosomes. Over time the endosome will, thus, mature from an early to a late endosome. During the course of maturation other processes besides recycling and sorting have to take place. Rab5 will be replaced by Rab7, and phosphatidylinositol 3-phosphate (PI3P) by phosphatidylinositol 3,5-bisphosphate $(PI3,5P_2)$ (Podinovskaia and Spang, 2018; Poteryaev et al, 2010; Wallroth and Haucke, 2018). In addition, endosomes need to acidify before fusion with lysosomes, and in many cells, endosomes move towards the cell center (Hu et al, 2015; Podinovskaia et al, 2021; Podinovskaia and Spang, 2018). This complex maturation process from early to late endosomes begs the question about the coordination between the individual processes. We have shown previously that while Rab11-dependent recycling can happen at any stage during endosome maturation, cross-talk between Rab conversion and acidification exists and Rab conversion has to be initiated before endosomes can fully acidify (Podinovskaia et al, 2021).

Whether Rab conversion and ILV formation are coordinated and if so, how remains unknown. Rab conversion is thought to be initiated by coincident detection of the Rab5GEF Rabex5 (RABX-5 in *Caenorhabditis elegans*) and increasing levels of PI3P on endosomes by the Rab7GEF Mon1/Ccz1 (Mon1 is called SAND-1 or CMON-1 in *C. elegans*) (Cabrera et al, 2014; Langemeyer et al, 2020; Poteryaev et al, 2010; Poteryaev et al, 2007). In addition, Rab5 can directly bind to Mon1/Ccz1 and modulate its activity (Borchers et al, 2023; Herrmann et al, 2023; Kinchen and Ravichandran, 2010; Langemeyer et al, 2020). These processes lead to the replacement of Rab5 by Rab7 on maturing endosomes (Borchers et al, 2021; Podinovskaia and Spang, 2018). Rab conversion takes only a few minutes in various experimental systems (Kinchen and Ravichandran, 2010; Podinovskaia et al, 2021; Poteryaev et al, 2010; Rink et al, 2005; Singh et al, 2014; Skjeldal et al, 2021; Yousefian et al, 2013). In parallel, ILV biogenesis promoted by the endosomal sorting complex required for transport (ESCRT) machinery has to occur (Appendix Fig. S1, Appendix Table S1). ESCRT-0 and -I are early acting ESCRT complexes that bind and select ubiquitinated cargo prone to degradation (Henne et al, 2011). The early ESCRTs corral the cargo towards the site of ILV formation (Cullen and Steinberg, 2018). There, cargo is deubiquitinated and transferred into ILVs, which are formed by the late acting ESCRTs ESCRT-III and -IV, in conjunction with auxiliary factors (Clague and Urbé, 2006; Henne et al, 2011) (Appendix Fig. S1, Appendix Table S1).

[1]Biozentrum, University of Basel, Basel, Switzerland. [2]Center for Microscopy and Image Analysis, University of Zurich, Zürich, Switzerland. ✉E-mail: anne.spang@unibas.ch

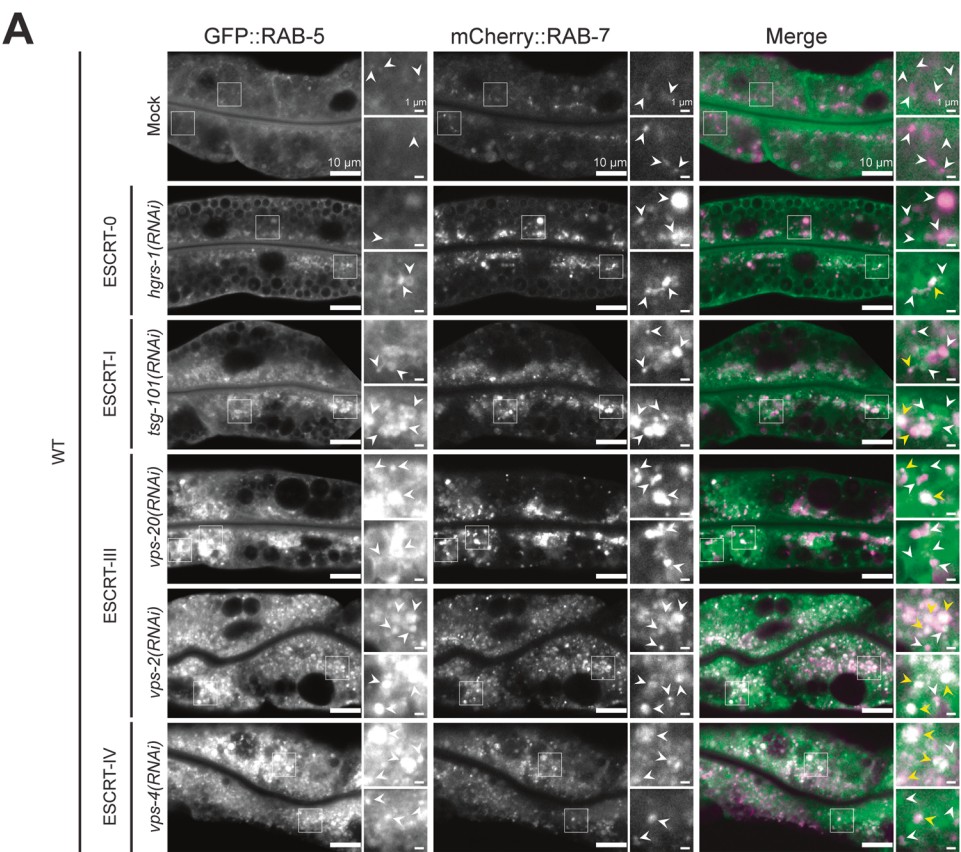

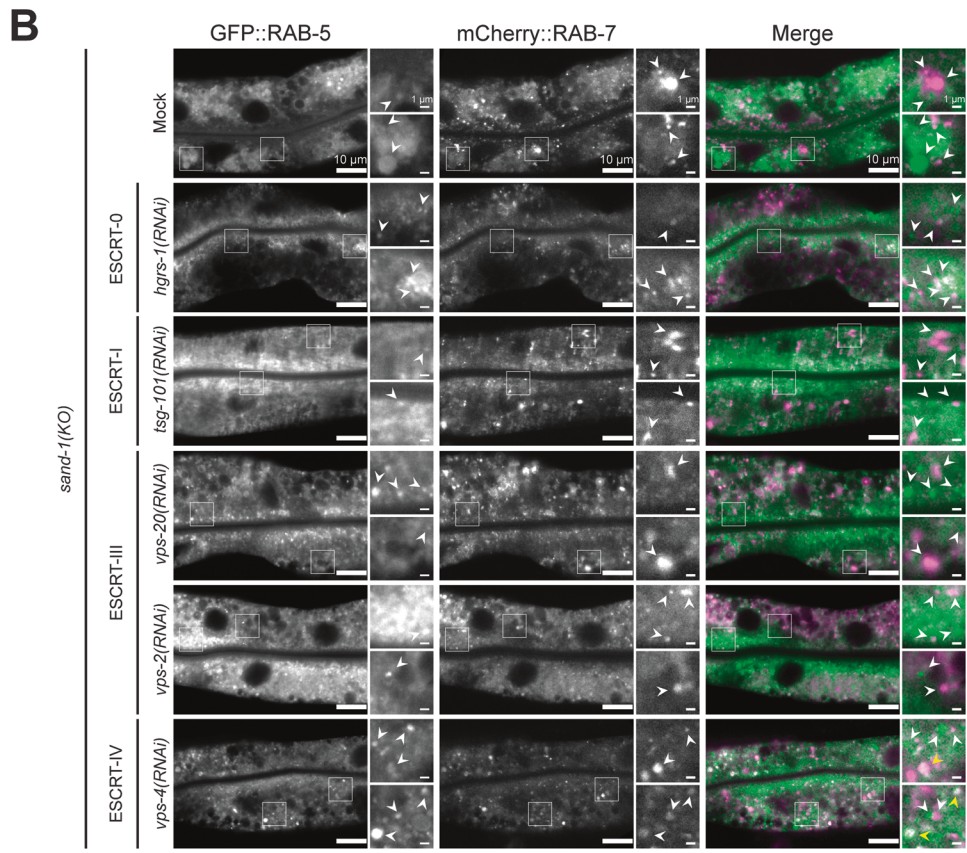

**Figure 1.   Knock-down of ESCRT mediated ILV formation impairs endosome maturation in WT and *sand-1(KO)*.**

(A) Knock-down of ESCRT components in a strain expressing GFP::RAB-5 and mCherry::RAB-7 in the intestine. Some of the individual signals are marked by white arrowheads. In the merged images, colocalization events are marked by yellow arrowheads (Signals: RAB-5 > green and RAB-7 > magenta). (B) Knockdown of ESCRT components in *sand-1(KO)* expressing GFP::RAB-5 and mCherry::RAB-7 in the intestine. White arrowheads pointing to GFP::RAB-5 and mCherry::RAB-7 positive structures, respectively in the individual channels. Colocalization events are marked by yellow arrowheads in the merges (Signals: RAB-5 > green and RAB-7 > magenta). Data information: Merged images were individually adjusted in all panels. Representative pictures with magnifications (white box) on the right are shown for each experiment (scale bars: 10 μm (main pictures) and 1 μm (insets)); $n = 3$ independent experiments. Source data are available online for this figure.

When sorting is completed the late endosome/multivesicular body (MVB) will fuse with a lysosome to form an endolysosome, in a process promoted by the homotypic fusion and protein sorting (HOPS) complex. Protein degradation takes place in the endolysosome (Huotari and Helenius, 2011; Spang, 2016; Szentgyörgyi and Spang, 2023).

At least the ESCRT-0 component Hrs (HGRS-1 in *C. elegans*) can already be detected on Rab5 positive endocytic vesicles (Pons et al, 2008; Raiborg et al, 2001; Solinger et al, 2022), indicating that cargo sorting into the degradative pathway is a very early event in the endosomal pathway. Whether and how the ESCRT machinery and Mon1/Ccz1 are coordinated remains unclear.

In this paper, we provide strong evidence for a connection between early ESCRTs, the recruitment of Mon1/Ccz1 to endosomes and Rab conversion in *C. elegans* and in mammalian cells. We show that early ESCRTs act upstream of Rab conversion and that loss of early ESCRTs prevents the recruitment of Rab7 onto endosomes. Likewise, overabundance of ubiquitinated cargoes block Rab conversion. We establish the necessity of a second Rab7GEF at least in the absence of Mon1/Ccz1 in vivo. We propose that the availability of ubiquitinated cargoes stabilizes the Rab5GEF Rabex5 on endosomes preventing its displacement by Mon1/Ccz1. Thus, corralling of cargoes towards sites of ILV formation by early ESCRTs is a key process in endosome maturation and a prerequisite for Rab conversion.

## Results

### A targeted RNAi screen reveals a potential connection between Rab conversion and ESCRT function

During endosome maturation a number of processes have to happen, at least some of which should be coordinated. To test whether ILV formation by the ESCRT complexes is coordinated with Rab5-to-Rab7 conversion, we performed an RNAi screen with ESCRT components in wild-type and *sand-1(KO)* mutant animals, in which Rab conversion is blocked (Poteryaev et al, 2010; Poteryaev et al, 2007; Solinger and Spang, 2014) (Figs. 1 and EV1). In both strains GFP-RAB-5 and mCherry-RAB-7 are expressed in the intestine, which only results in a small or modest increase of RAB protein abundance (Fig. EV4G). We observed genetic interactions, such as synthetic lethality, between a subset of ESCRT components and the Rab7GEF SAND-1, suggesting a link between Rab conversion and the ESCRT machinery (Appendix Tables S2 and S3). In wild-type animals, RAB-5 is mostly localized apically, close to the gut lumen with a few puncta distributed throughout the cell, while RAB-7 is mostly on apically localized endosomes, which are bit further away from the gut lumen

((Solinger and Spang, 2014), Fig. 1A). In contrast, in *sand-1(KO)* animals, RAB-5 endosomes are enlarged and the RAB-7 localization to apical endosomes is lost (Poteryaev et al, 2010; Poteryaev et al, 2007; Solinger and Spang, 2014) (Fig. 1B) As expected, knockdown of any of the ESCRT components affected the endosomal system, and RAB-5 positive endosomes were enlarged in wild-type animals when early ESCRTs, ESCRT-0 and -I, were depleted (Figs. 1A and EV1A). Some of the knock-downs showed conspicuously similar phenotypes to the ones observed in *sand-1(KO)* (Figs. 1 and EV1, Appendix Tables S4 and S5). However, in *sand-1(KO)* animals, knock-down of the early ESCRTs did not further increase the size of RAB5 positive endosomes (Figs. 1B and EV1B; Appendix Tables S4 and S5). Knock-down of late ESCRT components (ESCRT-III and ESCRT-IV) resulted in colocalization of RAB-5 and RAB-7 on endosomes in wild-type animals, indicating that Rab conversion could be initiated but not completed under these conditions. These data suggest that early ESCRTs may act upstream of Rab conversion, while late ESCRTs might act in parallel and downstream of Rab conversion. Moreover, since knock-down of early ESCRTs did not aggravate the *sand-1(KO)* mutant enlarged RAB5 positive endosome phenotype, it is conceivable that there is coordination between Rab conversion and ILV formation.

### The number of ILVs is reduced in ESCRT knock-downs and in *sand-1(KO)* animals

To gain a better understanding of the endosomal phenotypes, we performed transmission electron microscopy (TEM) on wild-type and *sand-1(KO)* animals, in which we also knocked down an early (TSG-101) or a late (VPS-2) ESCRT component (Figs. 2 and EV2; Appendix Fig. S2). Consistent with the live cell imaging and previous studies (Frankel et al, 2017; Poteryaev et al, 2010; Poteryaev et al, 2007; Poteryaev and Spang, 2005; Roudier et al, 2005; Solinger and Spang, 2014), endosomes were enlarged in *sand-1(KO)* and in an ESCRT-I knock-down, irrespective of the strain background or the tissue (Fig. EV2). In contrast ESCRT-III knockdown had a somewhat reduced effect on endosome size as also reported previously (Frankel et al, 2017). We defined an MVB in our micrographs as a structure that contained at least one ILV. As expected, the number of ILVs was decreased upon knock-down of ESCRT components (Fig. 2A,B). A similar effect was observed already in *sand-1(KO)* animals, but was not exacerbated by the concomitant knockdown of ESCRT components. Neither MVB nor ILV size was strongly affected in *sand-1(KO)* animals (Fig. 2C–F), consistent with the notion that the enlarged endosomes in *sand-1(KO)* animals are RAB-5 positive (Poteryaev et al, 2010; Poteryaev et al, 2007). Taken together, our data so far are consistent with a possible coordination between Rab conversion and ILV formation.

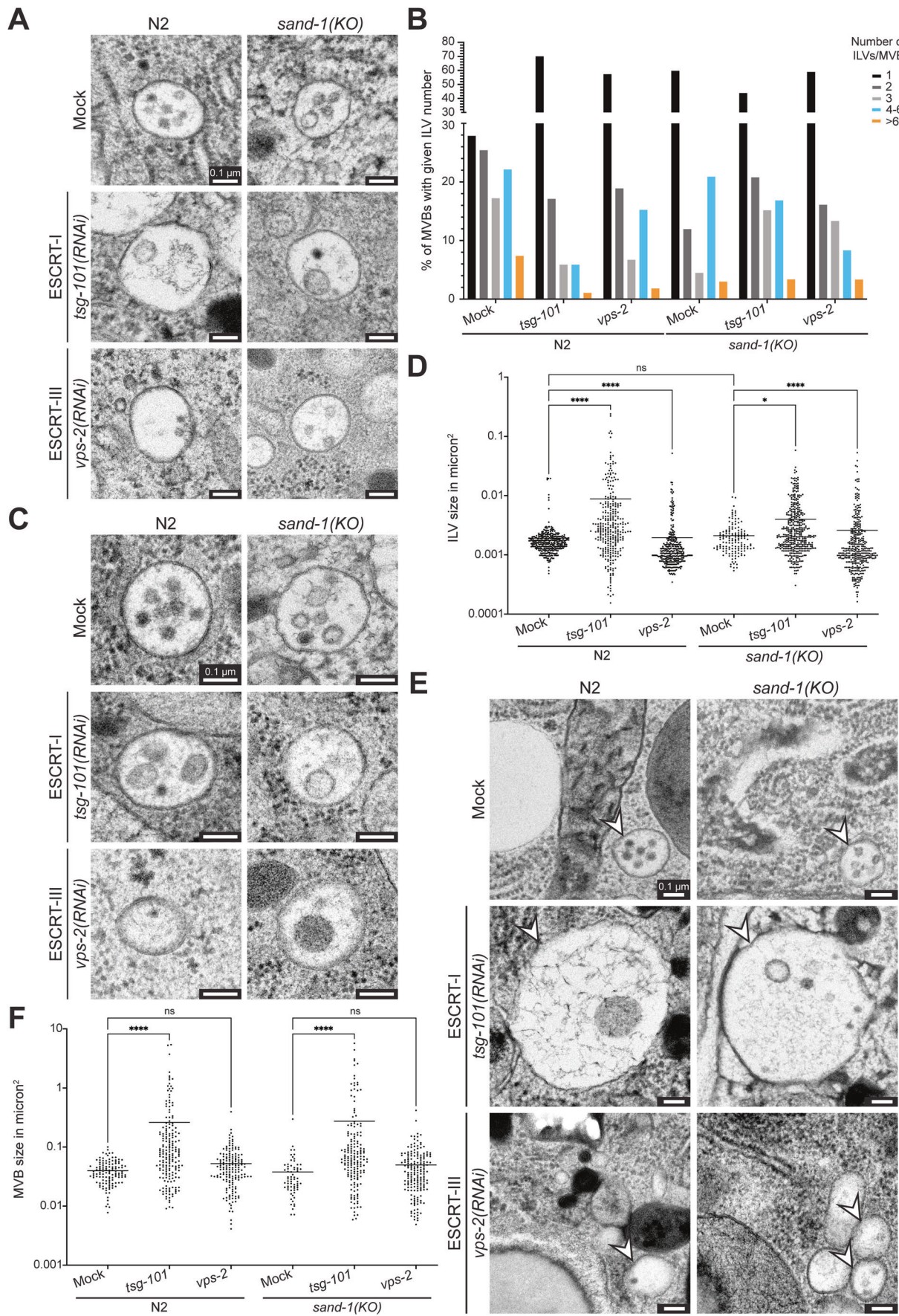

Figure 2.  ESCRT knockdowns affect the MVB and ILV size but not the ILV number in *sand-1 (KO)*.

(A) Knockdowns of *tsg-101* or *vps-2* reduce the number of ILVs in WT animals. *sand-1(KO)* reduces ILV number independently of ESCRT knockdown. (B) Quantification of the ILV phenotypes shown in (A) ILVs/MVB were counted in three worms per condition ($n = 3$) and summed in 5 categories, 1, 2, 3, 4-6 ILVs/MVB and more than 6 ILVs/MVB. The results per category are shown in % of the total amount as bar graph. (C) *tsg-101(RNAi)* causes in both strains an increase in ILV size. The *vps-2(RNAi)* resulted in the formation of smaller and bigger structures altering the ILV size range in both strains. *sand-1* knockout alone does not change the ILV size. (D) Quantification of the ILV size phenotypes depicted in (C). The size of each ILV was measured in $\mu m^2$ in three worm intestines per condition ($n = 3$). Data are shown in log 10 scale (Y-axis) with corresponding mean value as scatter plot. *P*-values: Mock vs. *tsg-101(RNAi)* $P = 2.4E-10$, Mock vs. *vps-2(RNAi)* $P = 4.8E-11$, Mock vs. *sand-1(KO)* $P > 0.9999$, *sand-1(KO)* vs. *sand-1(KO)* + *tsg-101(RNAi)* $P = 0.0439$, *sand-1(KO)* vs. *sand-1(KO)* + *vps-2(RNAi)* $P = 4.5E-5$. (E) MVB size phenotypes in WT and *sand-1(KO)* caused by *tsg-101(RNAi)* or *vps-2(RNAi)*. Knockdown of *tsg-101* but not of *vps-2* causes the formation of enlarged MVBs in both strains. The *sand-1* knockout alone does not change the MVB size. Representative MVBs are highlighted with white arrowheads in each picture. (F) The MVB size was quantified via measurement of the MVB area in $\mu m^2$ in three worm intestines per condition ($n = 3$). Displayed quantification belongs to (E). Data are shown in log 10 scale (Y-axis) with corresponding mean values as scatter plot. *P*-values: Mock vs. *tsg-101(RNAi)* $P = 5.8E-11$, Mock vs. *vps-2(RNAi)* $P > 0.9999$, Mock vs. *sand-1(KO)* $P > 0.9999$, *sand-1(KO)* vs. *sand-1(KO)* + *tsg-101(RNAi)* $P = 1.2E-9$, *sand-1(KO)* vs. *sand-1(KO)* + *vps-2(RNAi)* $P = 0.4844$. Data information: Pictures are individually adjusted (scale bars: 0.1 μm). Representative MVBs are shown for each examined condition (A, C and E). Kruskal–Wallis tests with Dunn's multiple comparisons tests were performed to determine the statistical significance between the examined conditions (D and F). Significance levels are displayed in the graphs as *$P \leq 0.05$, ****$P \leq 0.0001$ and ns $P > 0.05$. Unprocessed images and statistical raw data are available as source data. Source data are available online for this figure.

## HGRS-1 is present on RAB-5 positive and absent on RAB-7 positive endosomes

Our screen indicated that early ESCRTs act upstream of Rab conversion. Moreover, in mammalian cells the residence time of ESCRT-0 HRS (HGRS-1 in *C. elegans*) on endosomes is prolonged, when ESCRT-III components are depleted (Quinney et al, 2019). However, it remains unclear whether the HRS positive endosomes are Rab5 or Rab7 positive under these conditions. To address this issue, we knocked-down ESCRT-I, -III and -IV components in worms expressing GFP-HGRS-1 together with either RFP-RAB-5 or mCherry-RAB-7. HGRS-1 co-localized with RAB-5, even when late ESCRT components were depleted (Fig. 3A–D), while we did not observe colocalization between HGRS-1 and RAB-7 (Fig. 3E–G). Thus, the presence of HGRS-1 and RAB-7 on endosomes appears to be mutually exclusive, suggesting that HGRS-1 might have to leave endosomes prior to Rab conversion.

## The level of ubiquitinated cargo on endosomes impacts Rab conversion

The major role of ESCRT-0 is the corralling of ubiquitinated cargoes on early endosomes and hand them over to downstream ESCRTs for inclusion into ILVs. HRS binds directly to ubiquitinated cargo, and its presence on endosomes is strongly influenced by this function (Hirano et al, 2006; Komada et al, 1997; Raiborg et al, 2002; Wollert and Hurley, 2010). Accumulation of ubiquitinated cargoes would lead to persistent HRS localization on endosomes and thereby hamper endosomes maturation. To test this possibility, we first assessed whether ubiquitinated cargoes accumulate on endosomes when ESCRT levels are reduced by using a *C. elegans* strain expressing ubiquitin fused to GFP (GFP-UBQ) (Bakowski et al, 2014). Rather than looking at an individual cargo, we decided to look at ubiquitin on endosomes to more globally assess cargo effects. In control animals, GFP-UBQ is present in a haze with some occasional foci, presumably endosomes, in the cytoplasm and concentrated in the nucleus (Figs. 4A and EV3A). Knockdown of ESCRTs locked ubiquitin on endosomes and the nuclear pool was depleted (Figs. 4A,B and EV3A). Of note, depleting HGRS-1 or VPS-20 (ESCRT-III; CHMP6 in mammals) led to a reduction of the GFP-UBQ signal (Fig. EV3B), suggesting that some ubiquitin could be degraded under these conditions.

Moreover, the size of the GFP-UBQ positive endosomes was enlarged (Fig. EV3C). We also aimed to increase ubiquitinated cargo on endosomes by depleting the deubiquitinase USP-50, which removes ubiquitin from cargoes before they enter ILVs (Clague and Urbé, 2006; Henne et al, 2011). However, unfortunately, HGRS-1 is itself ubiquitinated (Katz et al, 2002; Polo et al, 2002; Stringer and Piper, 2011) and is also a substrate of USP-50 (Row et al, 2006; Zhang et al, 2014). Hence upon *usp-50(RNAi)*, HGRS-1 remains ubiquitinated and is degraded (Fig. EV3D). Still, our data suggest that reduction of ubiquitinated cargoes on endosomes may allow Rab conversion to occur. To test this hypothesis, we reduced the ubiquitin levels by RNAi in the *sand-1(KO)* mutant. We observed a decrease in size of the RAB-5 positive endosomes and RAB-7 was partially recruited to apically localized endosomes (Figs. 5A–C and EV4A,C). RAB-7 recruitment onto endosomes was sufficient to release the cargo hTFR-GFP from being trapped in internal vesicles (Fig. 5D–H) and also improved the morphology of the LMP-1 positive compartments (Figs. 5I–K and EV4B,C). Moreover, the colocalization of LMP-1 and RAB-7 remained high (Fig. 5L). Consistent with these findings, knockdown of *usp-50* in wild-type animals led to mislocalization of RAB-5 and colocalization of RAB-5 with RAB-7 and GFP-UBQ, respectively (Fig. EV5A–D; Appendix Tables S9 and S10). Furthermore, knockdowns of *hgrs-1* and *usp-50* showed negative genetic interaction with *sand-1(KO)* (Appendix Tables S2, S3, S6 and S7). Our data suggest that in the absence of ESCRT-0, or when ubiquitinated cargo cannot be corralled and accumulates on endosomes, endosomal flow is abrogated and Rab conversion is blocked. Moreover, relief from the accumulation of ubiquitinated cargo seemed to allow Rab conversion, even in the absence of the Rab7GEF SAND-1.

## VPS-39 is involved in the recruitment of a second Rab7GEF to endosomes

We showed above that reducing ubiquitination on endosomal cargo partially rescues the block in Rab conversion in the *sand-1(KO)* mutant. The SAND-1/CCZ-1 complex (Mon1/Ccz1 in mammals and yeast) is the GEF for RAB-7 (Nordmann et al, 2010). Therefore, another Rab7GEF should exist. Previously, the HOPS complex component Vps39 was proposed to act as an Rab7GEF (Binda et al, 2009; Wurmser et al, 2000), yet no strong Rab7GEF

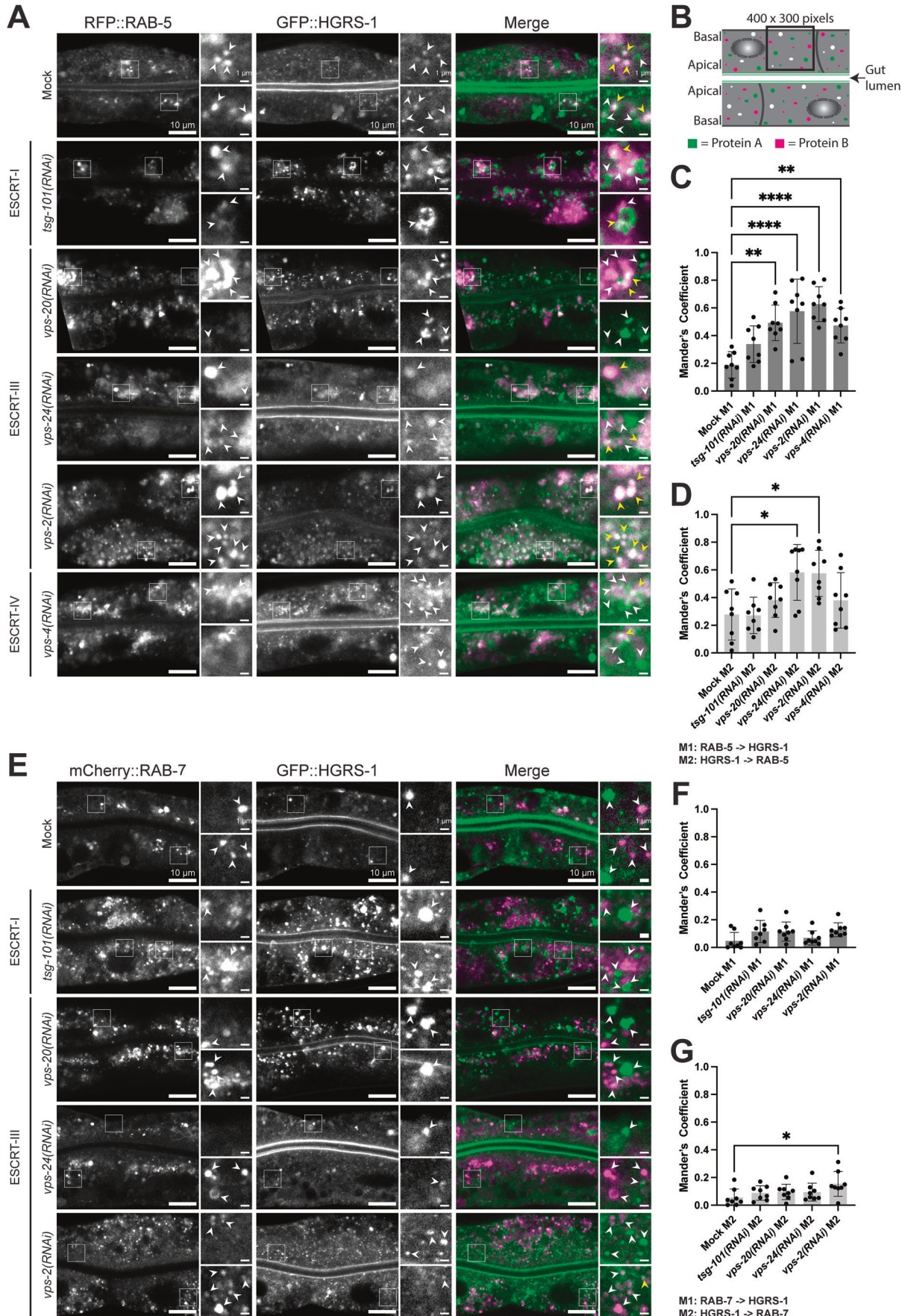

**Figure 3.   HGRS-1 and RAB-5 but not RAB-7 show high colocalization on endosomes if ILV formation is impaired.**

(A) GFP::HGRS-1 colocalizes with RFP::RAB-5 in early and late ESCRT factor knockdowns and is predominantly located on enlarged structures under *tsg-101*, *vps-20*, *vps-2*, and *vps-4* knockdown conditions. White arrowheads marking GFP::HGRS-1 and RFP::RAB-5 positive structures, respectively in the individual channels. Yellow arrowheads marking colocalization events in the merges (Signals: HGRS-1 > green and RAB-5 > magenta). (B) Schematical representation of a *C. elegans* gut with polarized intestinal cells and two expressed marked example proteins (Protein A in green and Protein B in magenta) which show partial colocalization (white). The colocalization quantification approach using a ROI (400 × 300 pixels) used in (C, D, F and G) is shown as an example. (C, D) Colocalization quantification of GFP::HGRS-1 and RFP::RAB-5 belonging to (A). For analysis shown in (C and D), 8 worms ($n = 8$) were examined per condition ($n = 3$ independent experiments). Analysis of colocalization via Mander's coefficient (M1: RFP::RAB-5 signal overlapping with GFP::HGRS-1 signal, M2: GFP::HGRS-1 signal overlapping with RFP::RAB-5 signal). (C) Obtained *P*-values: Mock M1 vs. *tsg-101(RNAi)* M1 $P = 0.3273$, Mock M1 vs. *vps-20(RNAi)* M1 $P = 0.002$, Mock M1 vs. *vps-24(RNAi)* M1 $P = = 5.6E-5$, Mock M1 vs. *vps-2(RNAi)* M1 $P = 5.5E-6$, Mock M1 vs. *vps-4(RNAi)* M1 $P = 0.0044$. (D) Obtained *P*-values: Mock M2 vs. *tsg-101(RNAi)* M2 $P > 0.9999$, Mock M2 vs. *vps-20(RNAi)* M2 $P = 0.8217$, Mock M2 vs. *vps-24(RNAi)* M2 $P = 0.0114$, Mock M2 vs. *vps-2(RNAi)* M2 $P = 0.0139$, Mock M2 vs. *vps-4(RNAi)* M2 $P = 0.8373$. (E) The very minor colocalization of GFP::HGRS-1 and mCherry::RAB-7 is only slightly affected under early or late ESCRT knockdown conditions and never reaches the levels of GFP::HGRS-1 and RFP::RAB-5 shown in (A). In the close ups white arrowheads marking GFP::HGRS-1 and mCherry::RAB-7 positive structures, respectively in the individual channels. Merges showing a combination of both channels (Signals: HGRS-1 > green and RAB-7 > magenta). Colocalization is indicated via yellow arrowhead. (F, G) Colocalization quantification of GFP::HGRS-1 and mCherry::RAB-7 belonging to (E). For analysis shown in (F and G), 8 worms ($n = 8$) were examined per condition ($n = 3$ independent experiments). Analysis of colocalization via Mander's coefficient (M1: mCherry::RAB-7 signal overlapping with GFP::HGRS-1 signal, M2: GFP::HGRS-1 signal overlapping with mCherry::RAB-7 signal). (F) Obtained *P*-values: Mock M1 vs. *tsg-101(RNAi)* M1 $P = 0.38$, Mock M1 vs. *vps-20(RNAi)* M1 $P = 0.3421$, Mock M1 vs. *vps-24(RNAi)* M1 $P > 0.9999$, Mock M1 vs. *vps-2(RNAi)* M1 $P = 0.0907$. (G) Obtained *P*-values: Mock M2 vs. *tsg-101(RNAi)* M2 $P > 0.9999$, Mock M2 vs. *vps-20(RNAi)* M2 $P > 0.9999$, Mock M2 vs. *vps-24(RNAi)* M2 $P > 0.9999$, Mock M2 vs. *vps-2(RNAi)* M2 $P = 0.0247$. Data information: Merges were individually adjusted in all panels. Representative pictures with enlargements (white box) on the right are shown for each experiment (scale bars: 10 μm (main pictures) and 1 μm (close ups)). Ordinary one-way ANOVA with Tukey's multiple comparisons test was performed for each experiment to determine the statistical significance between the examined conditions (C and D). For (F and G) a Kruskal–Wallis test with Dunn's multiple comparisons test was performed for each experiment for the same purpose. Significance levels are displayed in the graphs as *$P \leq 0.05$, **$P \leq 0.01$, ****$P \leq 0.0001$ and ns $P > 0.05$. Each data point in (C and D) and in (F and G) represents the colocalization of the markers measured in one worm. In the graphs (C, D, F and G) the mean ± s.d. is shown for all conditions. Unprocessed images and statistical raw data are available as source data. Source data are available online for this figure.

activity was detected in vitro (Nordmann et al, 2010). If VPS-39 was not a Rab7GEF, it might still be involved in the recruitment of another Rab7GEF. To test this possibility, we repeated the ubiquitin depletion in *sand-1(KO)*, but this time, we co-depleted VPS-39. Under these conditions, RAB-7 recruitment on apical endosomes was strongly reduced (Figs. 5A,B, 6A,B and EV4F). Thus, VPS-39 appears to be required for RAB-7 activation on endosomes in the absence of SAND-1. Moreover, the defect in endosome maturation in *sand-1(KO)* can be partially rescued by overexpression of RAB-7, as indicated by the LMP-1 localization (Figs. 5I,K and EV4B). Depletion of VPS-39 reversed this partial rescue and as a consequence, RAB-7 was dispersed and LMP-1 accumulated again (Figs. 6C,D, and EV4D,E). Moreover, the colocalization of RAB-7 and LMP-1 was significantly reduced in *vps-39(RNAi)* animals indicating that VPS-39 is required for efficient recruitment of RAB-7 to LMP-1 positive structures in *sand-1(KO)* (Fig. 6C,E). Finally, *vps-39(RNAi)* displayed a negative genetic interaction with *sand-1(KO)*, a phenotype which was not rescued by concomitant depletion of ubiquitin or overexpression of RAB-7 (Appendix Tables S6 and S7). Taken together, our data suggest that VPS-39 might be involved in the recruitment of a second Rab7GEF.

## RABX-5 prevents Rab conversion in the presence of ubiquitinated cargo

Next, we wanted to determine the link between the ubiquitinated cargoes, early ESCRTs and Rab conversion. Besides ESCRT-0, the GEF for RAB-5, RABX-5 (Rabex5 in mammals) has ubiquitin binding properties (Fig. 7A) (Dwivedi et al, 2011; Lee et al, 2006; Mattera et al, 2006; Penengo et al, 2006). In fact, together with the membrane-binding domain, the ubiquitin binding domain is required for the Rabex5 localization on early endosomes (Lauer et al, 2019; Mattera and Bonifacino, 2008; Mattera et al, 2006; Zhu et al, 2007). We have shown previously that SAND-1 interacts with and displaces RABX-5 from endosomes (Poteryaev et al, 2010). Thus, RABX-5 would be an ideal candidate to link the reduction of

ubiquitinated cargo to Rab conversion. We reasoned that as long as RABX-5 would be able to bind to ubiquitin, SAND-1 would be unable to displace it and hence, Rab conversion would be blocked. Once HGRS-1, together with other early ESCRTs, has corralled all ubiquitinated cargo into a domain for ILV formation, RABX-5 would no longer bind to ubiquitin and hence SAND-1 would be able to displace RABX-5 and initiate Rab conversion. To test this hypothesis, we knocked down RABX-5 in *sand-1(KO)* animals (Appendix Table S8). Similar to the knock-down of ubiquitin (Fig. 5), the RAB-7 localization to apical endosomes was restored (Fig. 7B,C). Moreover, the size and number of the RAB-5 positive endosomes was reduced (Fig. 7B,D,E). These results predicted that the concentration of endosomal ubiquitin influences RABX-5 localization. To test this prediction, we modulated the amount of ubiquitinated cargo on endosomes by either depleting ubiquitin (low abundance of ubiquitinated cargo) or HGRS-1 (high abundance ubiquitinated cargo) and determined the RABX-5 localization (Fig. 7F,G). Under low abundance of ubiquitinated cargo on endosomes, RABX-5 was mostly cytoplasmic, while under high abundance of ubiquitinated cargo, RABX-5 strongly accumulated on early endosomes. Reducing ubiquitin and HGRS-1 levels simultaneously gave a more severe RABX-5 mislocalization phenotype than the individual knockdowns, indicating a genetic interaction between ubiquitin and *hgrs-1* (Appendix Fig. S3B). Thus, our data indicate that RABX-5 is required for the coordination between early ESCRT function and Rab conversion by sensing uncorralled ubiquitinated cargoes on endosomes.

## The coordination between cargo corralling by ESCRT-0 and Rab conversion is conserved in mammalian cells

Finally, we wanted to test whether the crosstalk between early ESCRTs and Rab conversion is conserved in metazoans. First, we established that CRISPR-Cas9 knockout lines for ESCRT-0 (*HRS* KO) and ESCRT-III (*CHMP6* KO) led to the enlargement of Rab5 positive endosomes, like in the knockdown of the corresponding

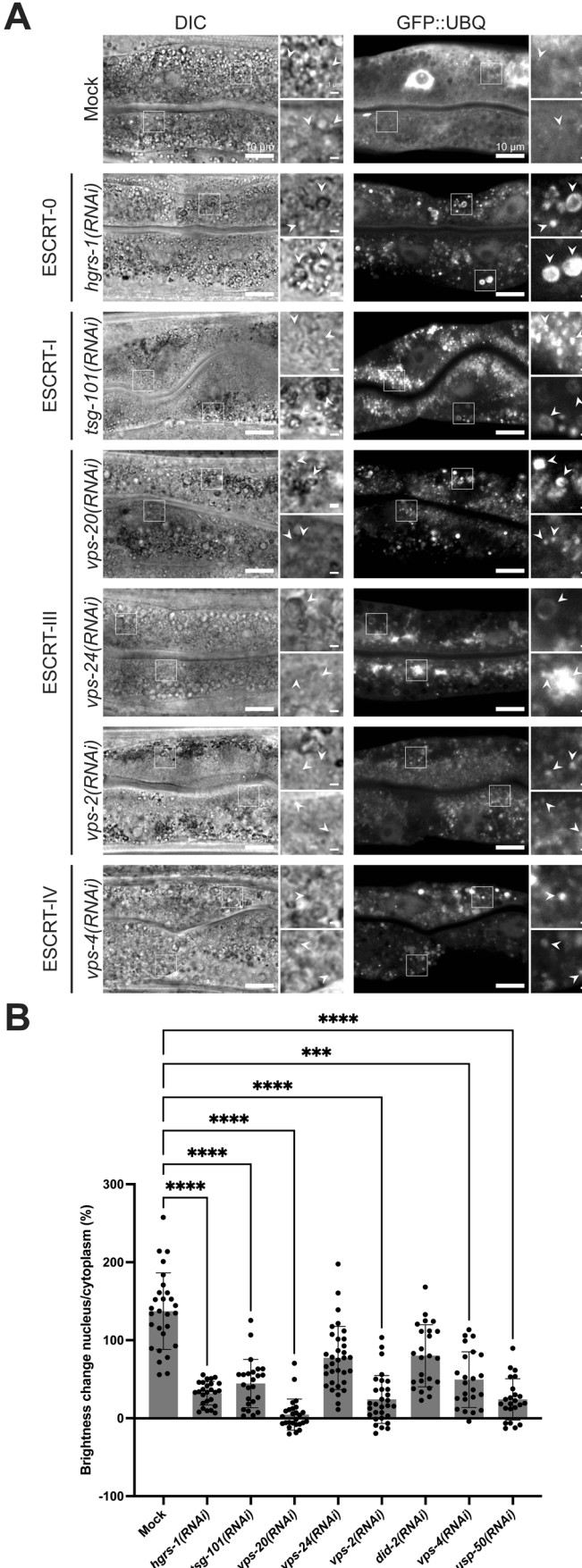

◄ **Figure 4.  Early and late ESCRT knockdowns affect UBQ localization in intestinal cells.**

(A) Knockdowns of core early ESCRT factors (0 and I) and late ESCRT factors (III–IV) cause a reduction in GFP::UBQ levels in the nucleus to varying degrees. White arrowheads indicating GFP::UBQ positive structures and the belonging region in the DIC. (B) Quantification of the nuclear GFP::UBQ amount with respect to the cytoplasmic signal in (A) and Fig. EV3A. Ten worms ($n = 10$) were quantified ($n = 3$ independent experiments). Each data point represents the result of one nucleus/cytoplasm comparison in the indicated condition. $P$-values: Mock vs. *hgrs-1(RNAi)* $P = 2.1\text{E}{-}8$, Mock vs. *tsg-101(RNAi)* $P = 2.3\text{E}{-}5$, Mock vs. *vps-20(RNAi)* $P < 1.0\text{E}{-}15$, Mock vs. *vps-24(RNAi)* $P = 0.1914$, Mock vs. *vps-2(RNAi)* $P = 3.0\text{E}{-}12$, Mock vs. *did-2(RNAi)* $P = 0.5219$, Mock vs. *vps-4(RNAi)* $P = 0.000113$, Mock vs. *usp-50(RNAi)* $P = 3.4\text{E}{-}10$. Data information: Shown are representative pictures with magnifications (white box) on the right for each experiment (scale bars: 10 μm (main pictures) and 1 μm (inlet)). Corresponding DIC pictures elucidate the extent of the gut, marker independent. GFP channel pictures of *hgrs-1* and *vps-20* knockdowns are increased in brightness (for analogous pictures with equal brightness see Fig. EV3B). Kruskal–Wallis test with Dunn's multiple comparisons test was performed for each experiment to determine the statistical significance between the examined conditions (B). Significance levels are displayed in the graphs as \*\*\*$P \le 0.001$, \*\*\*\*$P \le 0.0001$ and ns $P > 0.05$. The data show the mean ± s.d. for all examined conditions. Unprocessed images and statistical raw data are available as source data. Source data are available online for this figure.

proteins in *C. elegans* (Figs. 8A–C, 1A and 3A). Likewise, we confirmed that KO of the Rab7GEF component *CCZ1* displayed enlarged Rab5 positive endosomes (Fig. 8D,F) (Podinovskaia et al, 2021). Interestingly, similar to *C. elegans*, overexpression of Rab7 rescued the enlarged Rab5 positive endosomes phenotype in *CCZ1* KO cells (Figs. 8E–G and 5I), indicating that besides MON1/CCZ1 another Rab7GEF must exist also in mammalian cells. Moreover, when we overexpressed HRS in *CCZ1* KO cells, we rescued the enlarged Rab5 positive endosome phenotype (Fig. 8H–J). These data indicate that also in mammals HRS acts upstream of Rab conversion. Finally, we tested whether the interplay between ubiquitinated cargo and Rabex5 is conserved. As reported previously, mutating the ubiquitin binding sites in Rabex5 abolished Rabex5 localization to endosomes (Fig. 9A,B) (Mattera and Bonifacino, 2008; Mattera et al, 2006; Penengo et al, 2006). In contrast, overexpressing wild-type Rabex5 increased early endosome localization and endosome size. Taken together our data provide strong evidence for an evolutionary conserved mechanism by which ESCRT-0 function is required upstream of Rab conversion and that together with ubiquitinated cargo levels ESCRT-0 may serve as a timer for the initiation of Rab conversion.

## Discussion

How different processes during endosome maturation are coordinated is still poorly understood. Here, we investigated the interplay between Rab conversion and ILV biogenesis by the ESCRT machinery. We found that early ESCRT components, in particular HRS, act upstream of Rab conversion and that sorting of cargoes destined for degradation is a pre-requisite for the Rab5-to-Rab7 transition to occur. Our data indicate that ubiquitinated cargoes have to be sorted away from the place where Rab conversion can occur into domains that will give rise to ILVs on endosomes (Fig. 9C). This finding was initially surprising because Rab11-dependent sorting to the plasma membrane can occur from both Rab5 and Rab7 positive endosomes (Podinovskaia et al, 2021). However, the Rab5GEF Rabex5 binds to ubiquitin and hence as long as ubiquitinated cargo is still present, Rabex5 might be too stably bound to be displaced by the Rab7GEF Mon1/Ccz1 (Dwivedi et al, 2011; Lee et al, 2006; Mattera et al, 2006; Penengo et al, 2006; Poteryaev et al, 2010). A recent study in mammalian cells suggested that the MVB machinery scaffolding protein HD-PTP (PTPN23) and the Rabex5-binding protein Rabaptin5 are involved in the promotion of Rab conversion (Parkinson et al, 2021). Thus, we propose that successful corralling of ubiquitinated cargo is a pre-

requisite for Rab conversion (Fig. 9C). This model is supported by findings in yeast where deletion of either component of Rab7GEF Mon1/Ccz1 is synthetic lethal with loss of the yeast HRS homolog Vps27 (Aguilar et al, 2010; Costanzo et al, 2010; Costanzo et al, 2016; Surma et al, 2013). Likewise, Δ*ccz1* and Δ*vps9*, which encodes for yeast Rabex5 are synthetic lethal (Kucharczyk et al, 2009).

It seems as if slowly an order of events is emerging that governs endosome maturation. It appears as if sorting ubiquitinated cargoes into subdomains on endosomes that will give rise to ILVs is a pre-requisite for the initiation of Rab conversion. Thus, ubiquitinated cargo availability may determine the rate and/or timing at which endosomes can exchange their Rab proteins. We also know that Rab conversion is critical for endosome acidification (Podinovskaia et al, 2021). We still do not know what determines that early endosomes stop accepting new incoming cargo and initiate the maturation process (Huotari and Helenius, 2011; Maxfield and McGraw, 2004). Moreover, even though Rab11-dependent recycling can occur throughout the maturation process, it is likely that a signal exists, which indicates that recycling is completed. Such a signal could for example be the loss of the tubular part of the maturing endosomes or the loss of actin or sorting nexins. Also, it is clear that in a number of cellular systems, transport of the endosomes to the cell center is important for the maturation process (Huotari and Helenius, 2011; McDermott and Kim, 2015; Podinovskaia and Spang, 2018; Solinger and Spang, 2022). Again, how this transport is integrated into the overall maturation process still needs to be established.

We find that the HOPS complex, and in particular VPS-39, can recruit RAB-7 onto endosomes in the absence of SAND-1. In yeast, Vps39 was proposed previously to act as Rab7GEF (Wurmser et al, 2000). However, this possibility was largely dismissed by the identification of Mon1/Ccz1 as Rab7GEF (Nordmann et al, 2010). While our in vivo data would be consistent with a role of VPS-39 as Rab7GEF, the absence of Rab7GEF activity in vitro in the presence of the yeast HOPS complex rather suggest that the HOPS complex is involved in the recruitment of a second, yet unidentified Rab7GEF. We cannot, however, exclude the possibility that posttranslational modifications of the metazoan HOPS complex or the interaction with an auxiliary protein would turn on Rab7GEF activity. The HOPS complex can also be recruited to endosomes by Arl8 in Drosophila and mammalian cells (Khatter et al, 2015; Rosa-Ferreira et al, 2018; Schleinitz et al, 2023). Thus, it is formally possible that HOPS present on endosomes would recruit spontaneously activated Rab7 to endosomes. Given the intrinsically extremely low spontaneous exchange activity of Rab7, we consider this a rather unlikely possibility. We propose the existence of a

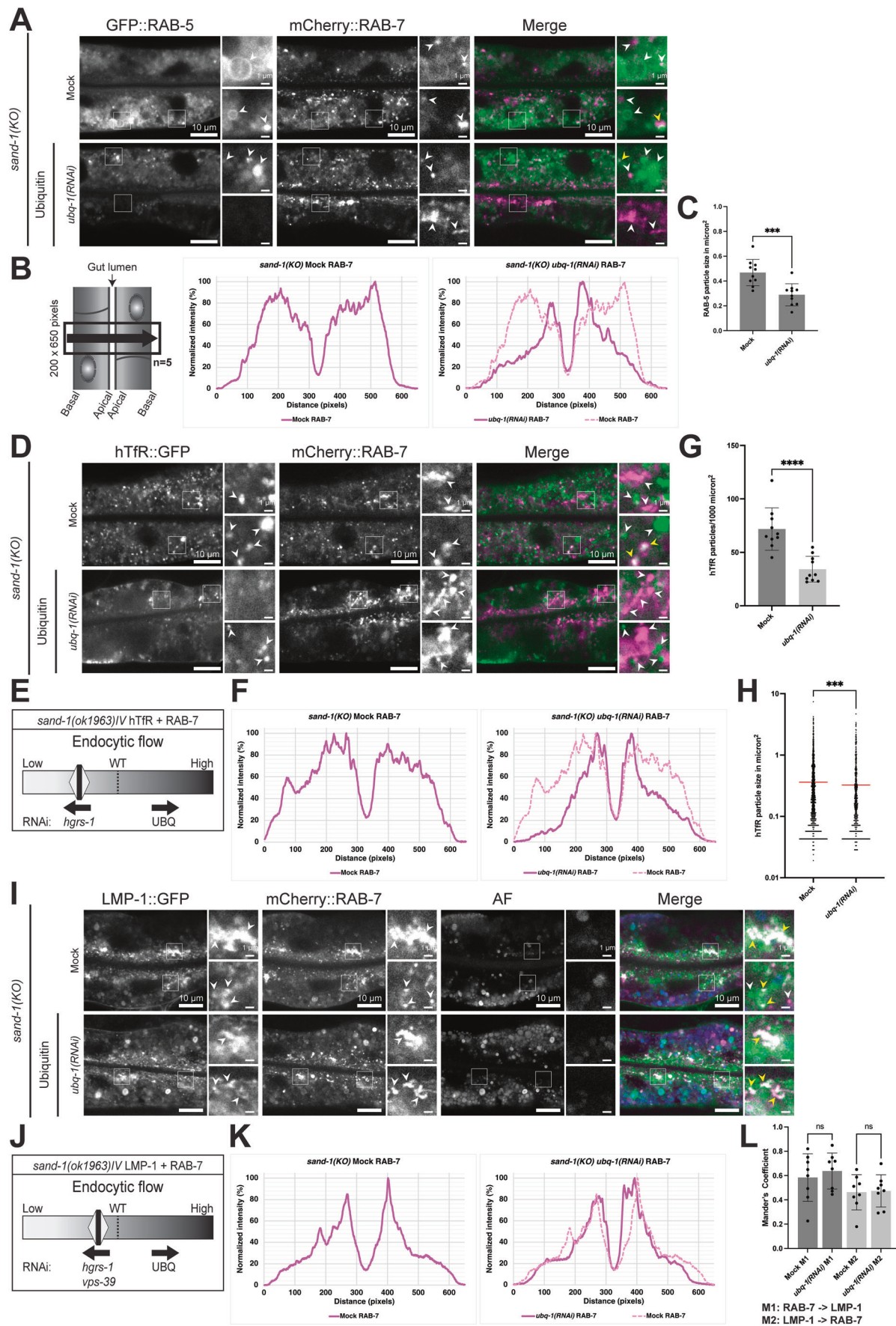

**Figure 5. The reduction of UBQ levels rescues RAB-7 endosome association in *sand-1(KO)*.**

(A) The reduction of UBQ levels in *C elegans* intestinal cells causes an accumulation of mCherry::RAB-7 on the apical side and reduces the size of GFP::RAB-5 positive structures in *sand-1(KO)*. White arrowheads point to GFP::RAB-5 and mCherry::RAB-7 positive structures, respectively in the individual channels. Colocalization events are highlighted with yellow arrowheads in the merges (Signals: RAB-5 > green and RAB-7 > magenta). (B, C) Quantification of the mCherry::RAB-7 area localization (B) and GFP::RAB-5 particle size in micron$^2$ (C) belonging to (A). For analyses shown in (B) 5 ($n = 5$), respectively 10 ($n = 10$) (C) worms were examined per condition ($n = 3$ independent experiments). In (C), each data point represents the mean particle size of one measured worm. *P*-value for (C): Mock vs. *ubq-1(RNAi)* $P = 0.000103$. (D) Reduction of UBQ levels in a *sand-1(KO)* mutant overexpressing an artificial endocytic cargo rescues mCherry::RAB-7 localization. The hTfR::GFP signal is shown with increased brightness in case of UBQ reduction condition to compensate for the reduced signal (for same brightness see Appendix Fig. S3A). White arrowheads pointing to hTfR::GFP and mCherry::RAB-7 positive structures, respectively in the individual channels. Merges show colocalization under Mock conditions with yellow arrowheads (Signals: hTfR > green and RAB-7 > magenta). (E) Schematic representation of the impaired endocytic flow in the *sand-1(KO)* overexpressing tagged hTfR and RAB-7. Effects of UBQ reduction and *hgrs-1(RNAi)* are indicated with black arrows. (F–H) mCherry::RAB-7 area localization quantification (F) and quantification of the hTfR::GFP particle abundance (per 1000 micron$^2$) (G) and size (in micron$^2$) (H) shown in (D). For analyses shown in (F) 5 ($n = 5$), (G) 10 ($n = 10$), and (H) 10 ($n = 10$) worms were examined per condition ($n = 3$ independent experiments). In (G) each data point represents the particle abundance of one examined worm and in (H) all measured particles of the 10 worms per condition are shown ($n > 2500$). (G) *P*-values: Mock vs. *ubq-1(RNAi)* $P = 4.3E{-}5$; (H) *P*-values: Mock vs. *ubq-1(RNAi)* $P = 0.000110$. (I) Reduction of UBQ levels, partially rescues the mCherry::RAB-7 localization in *sand-1(KO)*. White arrowheads point to LMP-1::GFP and mCherry::RAB-7 positive structures, respectively in the individual channels. The AF channel depicts autofluorescent gut granules, also visible in the GFP channel and the merge. In the merges colocalization is shown with yellow arrowheads (Signals: LMP-1 > green, RAB-7 > magenta and AF > blue). (J) Schematic representation of the endocytic flow in the *sand-1(KO)* strain overexpressing tagged LMP-1 and RAB-7. The consequences of the UBQ reduction and *hgrs-1(RNAi)* and *vps-39(RNAi)* are shown as black arrows. (K, L) Quantification of the mCherry::RAB-7 area localization (K) (equivalent quantification of LMP-1::GFP is shown in Fig. EV4B,C) and LMP-1::GFP/mCherry::RAB-7 colocalization (L) shown in (I). For analyses shown in (K) 5 ($n = 5$), respectively, 8 ($n = 8$) (L) worms were examined per condition ($n = 3$ independent experiments). (L) each data point represents the colocalization of the markers measured in one worm. Analysis of colocalization via Mander's coefficient (M1 mCherry::RAB-7 signal overlapping with LMP-1::GFP signal, M2: LMP-1::GFP signal overlapping with mCherry::RAB-7 signal). (L) *P*-values: Mock M1 vs. *ubq-1(RNAi)* M1 $P = 0.9923$, Mock M2 vs. *ubq-1(RNAi)* M2 $P > 0.9999$. Data information: Merged images were individually adjusted in all panels. Representative pictures with magnifications (white box) on the right are shown for each experiment (scale bars: 10 μm (main pictures) and 1 μm (insets)). Ordinary one-way ANOVA with Tukey's multiple comparisons test was performed for each experiment to determine the statistical significance between the examined conditions (C and L). For (G and H), a Mann–Whitney test (two-tailed) was performed for each experiment for the same purpose. Significance levels are displayed in the graphs as ***$P \leq 0.001$, ****$P \leq 0.0001$ and ns $P > 0.05$. The data show the mean ± s.d. for all examined conditions (C, G and L). In (H), the mean is shown for each condition as red line in the graph. Unprocessed images and statistical raw data are available as source data. Source data are available online for this figure.

second Rab7GEF. It was previously suggested that Mon1/Ccz1 is not able to recruit Rab7 onto lysosomes (Yasuda et al, 2016). Yet, the second GEF might be even recruited earlier. First, SAND-1 appears concomitant with RAB-7 on endosomes, but while SAND-1 disappears after about 2 min during Rab conversion, RAB-7 stays behind and its levels even increase afterwards (Poteryaev et al, 2010). Thus, Mon1/Ccz1 acts as the switch through coincidence detection of Rabex-5 and PI3P, while the other Rab7GEF activates Rab7 in a more sustainable manner. Moreover, the existence of a second Rab7GEF can also explain the observed bypass of *sand-1(KO)* caused by *vps-33.1 + 2(RNAi)* and the partial rescue of the Rab conversion deficiency in knockdowns of the FERARI subunits *spe-39* and *vps-45* (Solinger et al, 2020; Solinger and Spang, 2014). Finally, Mon1/Ccz1 binds PI3P but does not bind PI3,5P$_2$ efficiently (Poteryaev et al, 2010). It is tempting to speculate that similar events take place during autophagosome maturation before fusion with lysosomes.

# Methods

### Reagents and tools table

| Reagent/Resource | Reference or Source | Identifier or Catalog Number |
| --- | --- | --- |
| **Experimental models** | | |
| N2 Bristol (*C. elegans*) | CGC (Caenorhabditis Genetics Center)/Attila Stetak | N2 |
| *sand-1(ok1963)IV* (*C. elegans*) | CGC (Caenorhabditis Genetics Center) | RB1598 |
| *pwIs429[pvha-6::mCherry::rab-7]; pwIs72[pvha6::GFP::rab-5 + unc-119(+)]* (*C. elegans*) | Solinger and Spang, 2014 | FA086 |
| *sand-1(ok1963)IV; pwIs429[pvha-6::mCherry::rab-7]; pwIs72[pvha6::GFP::rab-5 + unc-119(+)]* (*C. elegans*) | Solinger and Spang, 2014 | AG Spang |
| *sand-1(ok1963)IV; pwIs429[vha-6::mCherry::rab-7]; pwIs90[Pvha-6::hTfR-GFP; Cbr-unc-119(+)]* (*C. elegans*) | Solinger and Spang, 2014 | AG Spang |
| *pwIs429[pvha-6::mCherry::rab-7] + pwIs50[Plmp-1::lmp-1::GFP + Cb-unc-119(+)]; sand-1(ok1963)IV* (*C. elegans*) | This study | AG Spang |
| *pwIs518[vha-6::GFP-HGRS-1]; pwIs846[Pvha-6-RFP-rab-5; Cb.unc-119(+)]* (*C. elegans*) | This study | AG Spang |
| *unc-119(ed3)III;pwIs429 [Pvha-6::mCherry::rab-7 + Cb-unc-119]; pwIs518[vha-6::GFP-HGRS-1]* (*C. elegans*) | This study | AG Spang |
| *jyEx128 [vha-6p:::GFP::UBQ, cb-unc-119(+)]; unc-119(ed3)III* (*C. elegans*) | Bakowski et al, 2014 | ERT261 |
| *unc-119(ed3)III; pwIs846[Pvha-6-RFP-rab-5; Cb.unc-119(+)]; jyEx128 [vha-6p:::GFP::UBQ, cb-unc-119(+)]; unc-119(ed3)III* (*C. elegans*) | This study | AG Spang |

| Reagent/Resource | Reference or Source | Identifier or Catalog Number |
|---|---|---|
| ycxls333[Pvha-6::GFP::RABX-5]; ycxEX1259[Pvha-6::mCherry::RAB-5] (C. elegans) | Zhang et al, 2020 | HUS5573 |
| HeLa CCL2 (H. sapiens) | ATCC (American Type Culture Collection) | RRID: CVCL_0030 |
| HeLa CCL2 HGS knockout (H. sapiens) | This study | AG Spang |
| HeLa CCL2 CHMP6 knockout (H. sapiens) | This study | AG Spang |
| HeLa CCL2 CCZ1 knockout (H. sapiens) | Podinovskaia et al, 2021 | AG Spang |
| HeLa CCL2 stably expression mApple-RAB5 and GFP-RAB7 (H. sapiens) | Podinovskaia et al, 2021 | AG Spang |
| **Recombinant DNA** | | |
| RNAi clones for C. elegans experiments | Kamath et al, 2003 | N/A |
| pDT7 tsg-101 (RNAi expression) | This study | N/A |
| pDT7 vps-2 (RNAi expression) | This study | N/A |
| pDT7 vps-60 (RNAi expression) | This study | N/A |
| pDT7 rabx-5 (RNAi expression) | This study | N/A |
| mApple Rab5 | Addgene | Cat #54944 |
| GFP Rab7 | Addgene | Cat #12605 |
| GFP Rab5 | Addgene | Cat #49888 |
| pCS2 HRS-RFP | Addgene | Cat #29685 |
| Rabex5 (myc-tagged) | Mattera and Bonifacino, 2008 | N/A |
| Rabex5 A58D (myc-tagged) | Mattera and Bonifacino, 2008 | N/A |
| Rabex5 Y25A/A58D (myc-tagged) | Mattera and Bonifacino, 2008 | N/A |
| **Antibodies** | | |
| Anti-RAB-5, Primary (polyclonal, generated in rabbit), generated in the Spang Lab, used at 1:500 (5% milk + TBS-T), for western blot | Poteryaev et al, 2007 | N/A |
| Anti-RAB-7, Primary (polyclonal, generated in rabbit), generated in the Spang Lab, used at 1:500 (5% milk + TBS-T), for western blot | Poteryaev et al, 2007 | N/A |
| Anti-α-Tubulin, Primary (monoclonal, generated with mouse cells), used at 1:10000 (5% milk + TBS-T), for western blot | Merck | Cat #T5168 |
| Anti-RAB5, Primary (-, originated from mouse), used at 1:1000 (5% milk + TBS-T), for western blot | Gift from Martin Spiess | N/A |

| Reagent/Resource | Reference or Source | Identifier or Catalog Number |
|---|---|---|
| Anti-RAB7, Primary (monoclonal, generated with rabbit cells), used at 1:1000 (Can Get Signal solution), for western blot | Cell Signaling Technology | Cat #9367 |
| Anti-C-MYC, Primary (monoclonal, generated with mouse cells), used at 1:200 (5% FBS in PBS), for immunofluorescence | Thermo Fisher Scientific | Cat #MA1-980 |
| Anti-Rabbit IgG (H + L), Secondary (polyclonal, generated in goat), used at 1:10,000 (TBS-T), for western blot | Thermo Fisher Scientific | Cat #31460 |
| Anti-Mouse IgG (H + L), Secondary (polyclonal, generated in goat), used at 1:10,000 (TBS-T), for western blot | Thermo Fisher Scientific | Cat #31430 |
| Anti-Mouse IgG (gamma 1), Secondary (polyclonal, generated in goat), used at 1:500 (5% FBS in PBS), for immunofluorescence | Thermo Fisher Scientific | Cat #A-21121 |
| **Oligonucleotides and other sequence-based reagents** | | |
| ESCRT-0, HGS, CRISPR primers for Exon1 and Exon22 | This study | Appendix Table S13 |
| ESCRT-III, CHMP6, CRISPR primers for Exon3 and Exon5 | This study | Appendix Table S13 |
| ESCRT-I, tsg-101, primers for cloning into RNAi vector pDT7 | This study | Appendix Table S15 |
| ESCRT-III, vps-2, primers for cloning into RNAi vector pDT7 | This study | Appendix Table S15 |
| ESCRT-III add., vps-60, primers for cloning into RNAi vector pDT7 | This study | Appendix Table S15 |
| RAB-5 GEF, rabx-5, primers for cloning into RNAi vector pDT7 | This study | Appendix Table S15 |
| **Chemicals, Enzymes and other reagents** | | |
| Tetramisole hydrochloride | Sigma-Aldrich | Cat #T1512-2G |
| Sodium azide | Carl Roth | Cat #K305.1 |
| Epon 812 | Sigma-Aldrich | Cat #45345 |
| Durcupan ACM | Sigma-Aldrich | Cat #44611 |
| Dibutyl phthalate | Sigma-Aldrich | Cat #524980 |
| DDSA | Sigma-Aldrich | Cat #45346 |
| DMP-30 | Sigma-Aldrich | Cat #45348 |
| DMEM (high-glucose) | Sigma-Aldrich | Cat #D1145-500ML |
| FBS (Batch: S00NC) | Biowest | Cat #S1810-500 |

| Reagent/Resource | Reference or Source | Identifier or Catalog Number |
|---|---|---|
| Penicillin-Streptomycin (solution stabilized) | Sigma-Aldrich | Cat #P4333-20ML |
| Fluoromount-G | SouthernBiotech | Cat #0100-01 |
| EcoRV-HF | New England Biolabs | Cat #R3195S |
| **Software** | | |
| Fiji/ImageJ2 | https://imagej.net/software/fiji; Rueden et al, 2017 | N/A |
| MultiStackReg | https://biii.eu/multistackreg; Brad Busse, National Institutes of Heatlh, USA | N/A |
| TurboReg | http://bigwww.epfl.ch/thevenaz/turboreg; Thévenaz et al, 1998 | N/A |
| JaCoP | https://imagej.net/plugins/jacop; Bolte and Cordelières, 2006 | N/A |
| Huygens Software | https://svi.nl/Huygens-Software; Scientific Volume Imaging | N/A |
| GraphPad Prism 9 | https://www.graphpad.com/scientific-software/prism; GraphPad Software | N/A |
| Microsoft Excel | https://www.microsoft.com/de-ch; Microsoft | N/A |
| Microsoft Word | https://www.microsoft.com/de-ch; Microsoft | N/A |
| Microsoft PowerPoint | https://www.microsoft.com/de-ch; Microsoft | N/A |
| OMERO | https://www.openmicroscopy.org/omero; University of Dundee & Open Microscopy Environment | N/A |
| Olympus FV31S-SW | https://www.olympus-lifescience.com/en; Olympus | N/A |
| Zeiss Zen Blue 2.6 | https://www.zeiss.com/microscopy/de/produkte/software/zeiss-zen.html; Carl Zeiss | N/A |
| Maps Offline Viewer | https://www.thermofisher.com/ch/en/home/global/forms/industrial/maps-offline-viewer-v3.html; Thermo Fisher Scientific | N/A |
| EndNote 21 | https://endnote.com; Clarivate | N/A |
| NEBuilder Assembly Tool | https://nebuilder.neb.com/#!; New England BioLabs | N/A |
| CHOPCHOP | https://chopchop.cbu.uib.no; Labun et al, 2019 | N/A |
| Adobe Illustrator 27.6.1 | https://www.adobe.com/ch_de/products/illustrator.html; Adobe | N/A |

| Reagent/Resource | Reference or Source | Identifier or Catalog Number |
|---|---|---|
| Fusion-CAPT (FUSION-FX7 advanced) | https://www.vilber.com; Vilber Lourmat | N/A |
| **Other** | | |
| Olympus FV3000 confocal laser scanning microscope | Olympus Schweiz | N/A |
| Zeiss LSM 880 confocal laser scanning microscope with Airyscan | Carl Zeiss | N/A |
| EM HPM100 high-pressure freezer | Leica Microsystems | N/A |
| AFS2 freeze-substitution unit | Leica Microsystems | N/A |
| Talos 120 transmission electron microscope | Thermo Fisher Scientific | N/A |
| Stemi 2000 stereomicroscope | Carl Zeiss | N/A |
| BD FACS ArialII Cell Sorter | Becton, Dickinson and Company | N/A |
| Axio Observer Zeiss microscope | Carl Zeiss | N/A |
| Fusion FX7 image acquisition system | Vilber Lourmat | N/A |

## Worm husbandry and general methods

*C. elegans* were grown and crossed according to standard methods (Brenner, 1974). All experiments and strains were grown at 20 °C. At this temperature, *sand-1(KO)* strains are still viable but show already severe endosomal trafficking phenotypes (Poteryaev et al, 2010; Poteryaev et al, 2007; Poteryaev and Spang, 2005; Solinger and Spang, 2014). The following *C. elegans* transgenes were used in this study: *sand-1(ok1963)IV, unc-119(ed3)III, pwIs72[pvha6::GFP::rab-5 + unc-119(+)], pwIs846[Pvha-6-RFP-rab-5; Cb.unc-119(+)], pwIs429[pvha-6::mCherry::rab-7], pwIs90[Pvha-6::hTfR-GFP; Cbr-unc-119(+)], pwIs50[Plmp-1::lmp-1::GFP + Cb-unc-119(+)], pwIs518[vha-6::GFP-HGRS-1], jyEx128 [vha-6p:::GFP::UBQ, cb-unc-119(+)], ycxIs333[pvha6::GFP::rabx-5]* and *ycxEx1259[pvha-6::mCherry::rab-5]*. Strains used in this study are listed in Appendix Table S11 and the Reagents and Tools Table.

## RNAi by feeding in *C. elegans*

For each experiment 5 L4/young adult animals were transferred on an isopropyl β-D-1-thiogalactopyranoside (IPTG) containing standard nematode growth medium (NGM) agar plate seeded with dsRNA producing *E. coli*. In case of pre-feeding experiments, several worms were transferred first on IPTG plates seeded with control dsRNAi expressing bacteria. L3 worms of the next generation (F1) were transferred on plates for RNAi and imaged when they reached young adulthood. Several plates per condition were set up in parallel as technical replicates. RNAi clones were either taken from the Ahringer library (Kamath et al, 2003) or created via cloning (see plasmid generation for *C. elegans*). All used

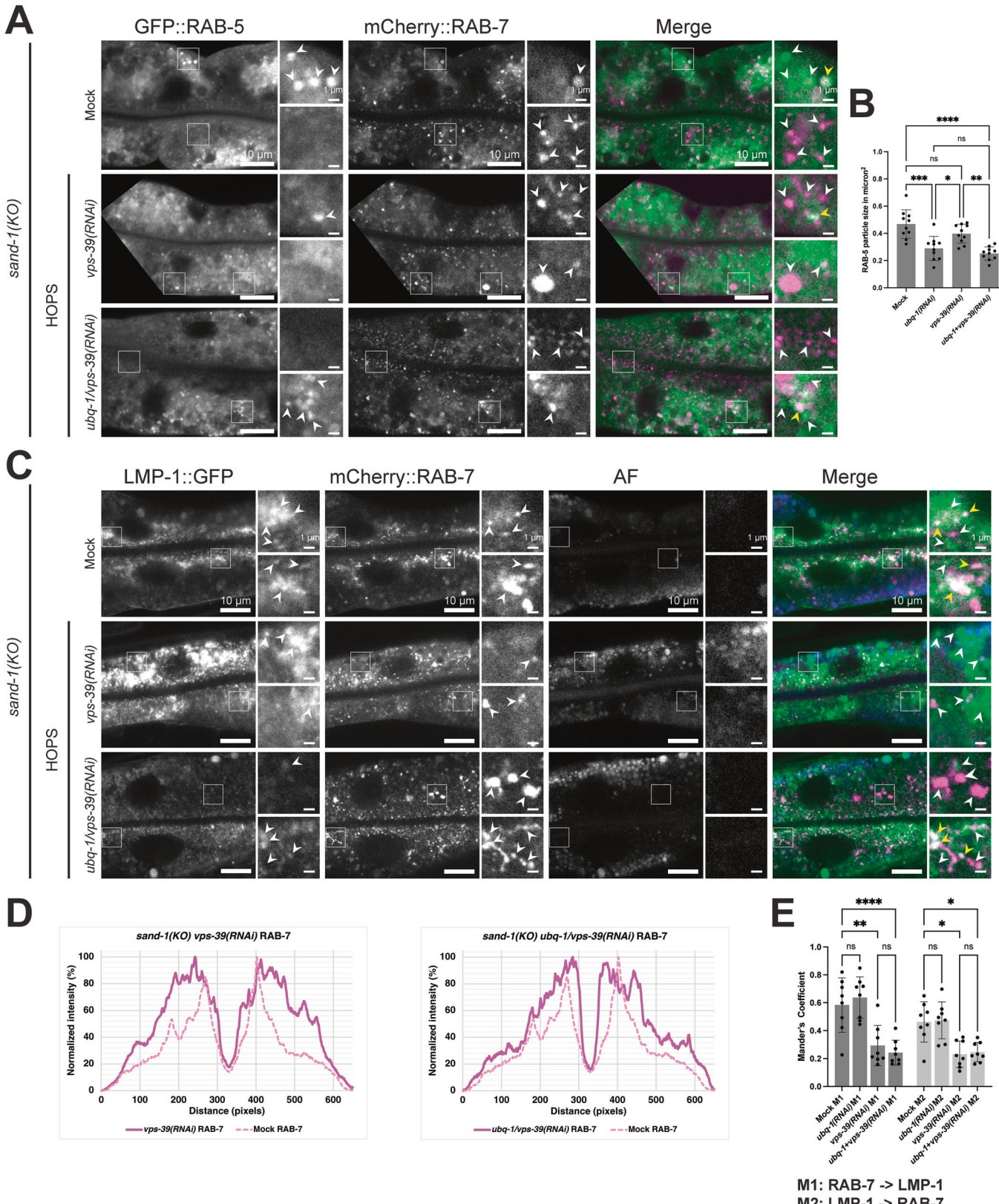

M1: RAB-7 -> LMP-1
M2: LMP-1 -> RAB-7

**Figure 6. VPS-39 facilitates RAB-7 recruitment to endosomes in *sand-1(KO)*.**

(A) The knockdown of *vps-39* abrogates the rescue of UBQ reduction in *sand-1(KO)* animals. White arrowheads point to GFP::RAB-5 and mCherry::RAB-7 positive structures, respectively in the individual channels. Colocalization events are marked with yellow arrowheads in the merges (Signals: RAB-5 > green and RAB-7 > magenta). (B) Quantification of the GFP::RAB-5 particle size in micron$^2$ belonging to (A). Control group and *ubq-1* knockdown are already shown in Fig. 5C and experimental details are the same like explained in the figure legend belonging to Fig. 5C ($n = 10$). *P*-values: Mock vs. *vps-39(RNAi)* $P = 0.2302$, Mock vs. *ubq-1+vps-39(RNAi)* $P = 4.4E-6$, *ubq-1(RNAi)* vs. *vps-39(RNAi)* $P = 0.0240$, *ubq-1(RNAi)* vs. *ubq-1+vps-39(RNAi)* $P = 0.7328$, *vps-39(RNAi)* vs. *ubq-1+vps-39(RNAi)* $P = 0.0015$. (C) In *vps-39* knockdown cells, LMP-1::GFP forms accumulations and mCherry::RAB-7 a dispersed pattern. These patterns are only partial rescuable via UBQ reduction in this *sand-1(KO)*. White arrowheads pointing to LMP-1::GFP and mCherry::RAB-7 positive structures, respectively in the individual channels. In addition, the AF channel depicts the gut granules also visible in the GFP channel. Colocalization events are marked with yellow arrowheads in the merges (Signals: LMP-1 > green, AF > blue and RAB-7 > magenta). (D) Quantification of the mCherry::RAB-7 area localization (equivalent quantification of LMP-1::GFP is shown in Fig. EV4D,E) belonging to (C). The control group is the same as shown in Fig. 5L and the data were collected and analyzed like described there. (E) Colocalization quantification of LMP-1::GFP and mCherry::RAB-7 belonging to (C). Control group and *ubq-1* knockdown as displayed in Fig. 5L and experimental details are visible in the figure legend Fig. 5L ($n = 8$). *P*-values: Mock M1 vs. *vps-39(RNAi)* M1 $P = 0.0013$, Mock M1 vs. *ubq-1+vps-39(RNAi)* M1 $P = 9.5E-5$, *vps-39(RNAi)* M1 vs. *ubq-1+vps-39(RNAi)* M1 $P = 0.9941$, Mock M2 vs. *vps-39(RNAi)* M2 $P = 0.0198$, Mock M2 vs. *ubq-1+vps-39(RNAi)* M2 $P = 0.0377$, *vps-39(RNAi)* M2 vs. *ubq-1+vps-39(RNAi)* M2 $P > 0.9999$. Data information: Merges were individually adjusted in all panels. Representative pictures with magnification (white box) on the right are shown for each experiment (scale bars: 10 µm (main pictures) and 1 µm (insets)). A detailed description of the performed statistics for (B and E) is available in the figure legend of Fig. 5. Significance levels are displayed in the graphs as *$P \leq 0.05$, **$P \leq 0.01$, ***$P \leq 0.001$, ****$P \leq 0.0001$ and ns $P > 0.05$. The data show the mean ± s.d. for all examined conditions (B and E). Unprocessed images and statistical raw data are available as source data. Source data are available online for this figure.

constructs were confirmed via sequencing (Microsynth AG, Balgach, Switzerland).

## Microscopy of *C. elegans*

Worms were imaged with an Olympus FV3000 confocal laser scanning microscope (Olympus Schweiz AG, Wallisellen, Switzerland) unless stated otherwise. The intensities of the 405, 488, and 561 nm lasers varied between 0.8 and 10%. For high resolution pictures, the Galvano scanner was used. The sampling speed was set to 8.0 µs/pixel and the scan direction was one-way scanning. Pictures were taken with a UPLSAPO60XS2 objective lens and silicone immersion oil (Olympus Schweiz AG). For image acquisition and processing at the microscope the FV31S-SW software (Olympus Corp, Tokyo, Japan) was used. Prior to imaging, worms were transferred from the plate into a drop of 33 mM Levamisole (0.5 ml Levamisole (100 mM) + 1 ml M9) on a 2% agarose pad on a slide. Coverslips were sealed with vaseline.

The images of the *rabx-5* knockdown in *sand-1(KO)* were obtained on a Zeiss LSM 880 with Airyscan (Carl Zeiss AG, Feldbach, Switzerland), using 488 nm and 555 nm lasers. The color channels were imaged in separate phases to reduce bleed through. Pictures were taken with a Plan-Apochromat 63x/1.4 Oil DIC M27 objective lens and immersion oil (Carl Zeiss AG) using the Airyscan Fast 3D mode with a resolution of 11.73 pixel per micron (Voxel size $0.0852 \times 0.0852 \times 0.1867$ µ$^3$). All images were taken and processed at the microscope using the Zen Blue software (Carl Zeiss AG, Oberkochen, Germany). Worms were paralyzed with 20 mM NaN$_3$.

## Transmission electron microscopy (TEM)

For TEM, worms were frozen as follows. *C. elegans* animals were picked from an agar plate and transferred to a droplet of M9 medium on a 100 µm cavity of a 3 mm aluminum specimen carrier (Engineering office M. Wohlwend GmbH, Sennwald, Switzerland). 5–10 worms were added to the droplet and the excess M9 medium was sucked off with dental filter tips. A flat aluminum specimen carrier was dipped in 1-hexadecene and added on top. Immediately, the specimen carrier sandwich was transferred to the middle plate of an EM HPM100 high-pressure freezer

(Leica Microsystems GmbH, Vienna, Austria) and frozen immediately.

Freeze-substitution was carried out in an AFS2 freeze-substitution unit (Leica Microsystems GmbH) with an integrated, custom built agitation system in water-free acetone containing 1% OsO$_4$ for 8 h at $-90$ °C, 7 h at $-60$ °C, 5 h at $-30$ °C, 1 h at 0 °C. Transition gradients of 30 °C/h, followed by 30 min incubation at RT were employed. Samples were rinsed twice with acetone water-free, block-stained with 1% uranyl acetate in acetone (stock solution: 20% in methanol) for 1 h at 4 °C, rinsed twice with water-free acetone and embedded in Epon-Araldite (Merck KGaA, Darmstadt, Germany) (composition in Appendix method section) with the following steps: 66% in acetone overnight, 100% for 1 h at RT and polymerized at 60 °C for 28 h. Ultrathin sections (50 nm) were transferred on one hole grids (with a Formvar film and coated with 15 nm of carbon) and post-stained with Reynold's lead citrate (composition in Appendix method section). Image acquisition was performed in a Talos 120 transmission electron microscope at 120 kV acceleration voltage equipped with a bottom mounted Ceta camera (Thermo Fisher Scientific FEI Europe B.V., Eindhoven, Netherlands) using the Maps software (Thermo Fisher Scientific Inc., Waltham, USA). Image analysis was performed using Maps software, Fiji and Prism (GraphPad Software LLC., Boston, USA).

## *C. elegans* MVB and ILV quantifications

To quantify the MVB and ILV phenotypes in *C. elegans* intestinal cells, unstitched high resolution (original pixel size = 8.2169 Å) TEM tile sets showing ultrathin sections of whole worm intestines were examined via Maps software (Thermo Fisher Scientific Inc.) to detect all MVBs (per worm one section was analyzed). In the next step, MVB and ILV size were recorded using Fiji, and the number of ILVs was counted manually (see Appendix Fig. S2). Statistical analyses and visualizations were performed in Prism (GraphPad Software LLC.).

## *C. elegans* population growth analysis

The growth of the worm populations during the RNAi experiments was evaluated using a Stemi 2000 stereomicroscope equipped with a KL 200 light source (Carl Zeiss AG). Once the F1 animals of the

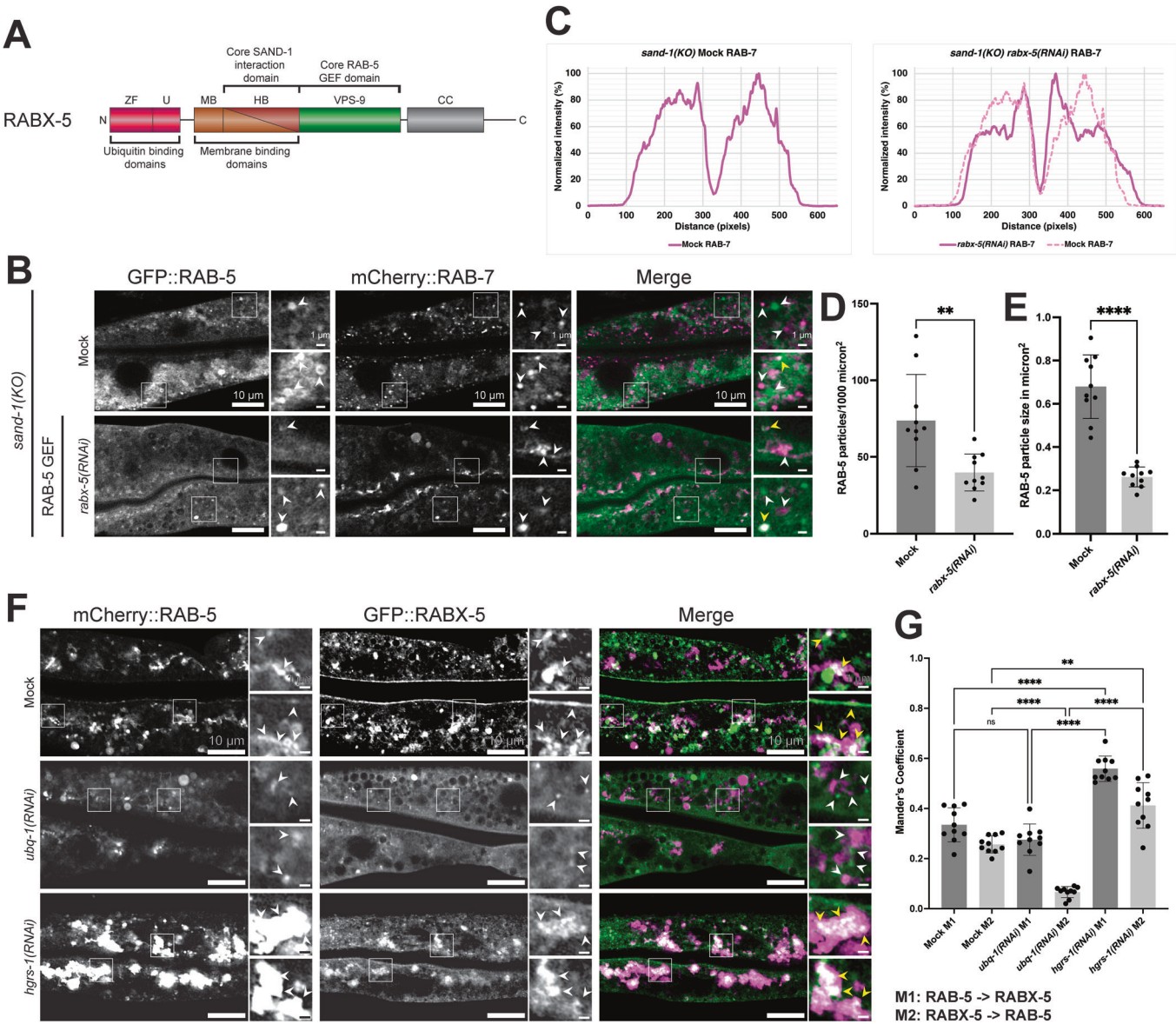

RNAi control reached young-adult stage, the developmental and growth phenotypes of the RNAi worms were determined. Each experiment was performed three times independently.

## Area plot quantification of RAB-7 and LMP-1 localization in the *C. elegans* intestine

The area localization quantification in the intestine was performed manually via Fiji. Middle planes of the gut (z-dimension) were used for the analysis. To measure the area localization a ROI (200 × 650 pixels) was placed in the worm on a representative spot covering gut and lumen horizontally. A line measurement using "plot profile" was performed. This measurement generates a line plot consisting of 650 values describing the average brightness of a 200-pixel long line each (see Fig. 5B and Fig. EV4A). In case of a multichannel image the same ROI was placed on the equivalent spot in the second channel and the measurement was repeated in

the same way. The same ROI was used for each worm throughout the whole quantification to be consistent. The brightness values of each worm were normalized to 100, as were the final mean values.

## Quantification of number and size of RAB-5 or hTfR positive endosomes in *C. elegans*

The size of RAB-5 or hTfR positive structures in the worm gut was quantified using Fiji. For analysis one slice of the z-stack showing the middle plane of the worm gut (z-dimension) was selected. The picture was converted to binary and treated with the "open" and "watershed" functions to reduce background and sharpen structures. The remaining structures were then measured via the "analyze particles" function to obtain the area value of each particle. The number of RAB-5 or hTfR positive structures in the worm gut was counted and correlated with the gut area. Statistical analyzes and plotting were performed in Prism (GraphPad Software LLC.).

**Figure 7.  The endosomal association of RABX-5 is highly UBQ dependent and its depletion rescues *sand-1(KO)*.**

(A) Graphical illustration of *C. elegans* RABX-5. RABX-5 harbors following domains from N- to C-terminus required for its function: UBQ binding > zinc finger motif (ZF) and UBQ interacting motif (U) (slight cardinal red), Membrane binding > membrane-binding domain (MB) (slight brown) and helical bundle (HB) (dark brown), Core SAND-1 interaction > HB, Core RAB-5 GEF > VPS-9 domain (dark green), Autoinhibition > coiled coil (CC) domain (dark gray). (B) The knockdown of *rabx-5* causes an accumulation of GFP::RAB-5 and mCherry::RAB-7 in the apical area of the worm gut and a reduction in size and abundance of GFP::RAB-5 structures in *sand-1(KO)*. White arrowheads pointing to GFP::RAB-5 and mCherry::RAB-7 positive structures, respectively in the individual channels. Colocalization events are marked with yellow arrowheads in the merges (Signals: RAB-5 > green and RAB-7 > magenta). (C–E) Quantification of the mCherry::RAB-7 area localization (C), GFP::RAB-5 particle abundance (per 1000 micron$^2$) (D) and size (in micron$^2$) (E) belonging to (B). For analysis shown in (C) 5 ($n = 5$), (D) 10 ($n = 10$) and (E) 10 ($n = 10$) worms were examined per condition ($n = 3$ independent experiments). In (D) each data point represents the particle abundance of one examined worm and in (E) each data point shows the mean particle size of one measured worm. (D) P-value: Mock vs. *rabx-5(RNAi)* $P = 0.0063$; (E) P-value: Mock vs. *rabx-5(RNAi)* $P = 3.9E-6$. (F) The knockdown of *ubq-1* causes a reduction of GFP::RABX-5 in the apical area of the worm gut. Conversely, the knockdown of *hgrs-1* causes an accumulation of mCherry::RAB-5 and increased colocalization of GFP::RABX-5 on these enlarged structures. White arrowheads pointing to mCherry::RAB-5 and GFP::RABX-5 positive structures, respectively in the individual channels. Colocalization events are marked with yellow arrowheads in the merges. (G) Quantification of the colocalization of mCherry::RAB-5 and GFP::RABX-5 in (F) by Mander's coefficient ($n = 10$ worms were measured). The Mander's coefficient are defined as follows; M1: mCherry::RAB-5 signal overlapping with GFP::RABX-5 signal and M2: GFP::RABX-5 signal overlapping with mCherry::RAB-5 signal. Each point represents one examined worm. Obtained P-values: Mock M1 vs.*ubq-1(RNAi)* M1 $P = 0.5110$; Mock M1 vs. *hgrs-1(RNAi)* M1 $P = 3.1E-6$; *ubq-1(RNAi)* M1 vs. *hgrs-1(RNAi)* M1 $P = 4.9E-8$; Mock M2 vs. *ubq-1(RNAi)* M2 $P = 1.4E-9$; Mock M2 vs. *hgrs-1(RNAi)* M2 $P = 0.0049$; *ubq-1(RNAi)* M2 vs. *hgrs-1(RNAi)* M2 $P = 5.2E-6$. Data information: Merges were individually adjusted in all panels. Representative pictures with magnification (white box) on the right are shown for each experiment (scale bars: 10 μm (main pictures) and 1 μm (insets)). Welch's t test (two-tailed) was performed for each experiment to determine the statistical significance between the examined conditions (D and E). For (G), a Brown–Forsythe ANOVA test and Welch's ANOVA test, followed by Dunnett's T3 multiple comparisons test were performed for the same purpose. Significance levels are displayed in the graphs as **$P \leq 0.01$, ****$P \leq 0.0001$ and ns $P > 0.05$. The data show the mean ± s.d. for all examined conditions (D, E and G). Unprocessed images and statistical raw data are available as source data. Source data are available online for this figure.

## Colocalization examination of RAB-5, RAB-7, HGRS-1, LMP-1, and RABX-5

Colocalization of two markers in *C. elegans* intestinal cells was quantified using Fiji. The two relevant channels were aligned manually or computer assisted (Fiji plugins TurboReg and Multi-StackReg (see Appendix Table S17)). In these pictures a ROI (400 × 300 pixels) was drawn and the colocalization of the two markers in these ROIs were measured (see Fig. 3B) with the Fiji plugin JACoP (see Appendix Table S17). The thresholds to calculate the Mander's coefficients (M1 and M2) were adjusted in each picture and for each channel individually to measure representative signals for each marker. Statistical analyzes and visualization were performed in Prism (GraphPad Software LLC.).

## Quantification of UBQ localization and structure size in *C. elegans*

The nuclear UBQ accumulation, cytoplasmic UBQ and the size of UBQ positive structures in *C. elegans* intestinal cells were measured manually using Fiji. A ROI (50 × 50 pixels) was put into the nucleus and a same size ROI next to nucleus in the cytoplasm. The fluorescence intensity of both ROIs was determined. The size of the UBQ positive structures was measured in the selected images with the "analyze particles" function, as described in "quantification of number and size of RAB-5 positive endosomes in *C. elegans*"; omitting the nucleus from the analyzes. Statistical analyzes were performed in Prism (GraphPad Software LLC.).

## Western blot analysis

Worms from 8 plates (mixed stage) were collected and lysed in FERARI buffer (50 mM HEPES pH 7.7, 10% glycerol, 150 mM KOAc, 2 mM MgCl$_2$ 1% NP-40) containing Halt protease inhibitor cocktail (Thermo Fisher Scientific Inc.). 10–20 μg of total protein was loaded onto 8–17% SDS–PAGE gradient gels before transfer onto nitrocellulose membranes (Amersham Protran; 10600003).

Membranes were blocked with 5% milk, 0.1% Tween 20 for 1 h at room temperature. The primary antibody incubation was overnight at 4 °C and the secondary horseradish peroxidase (HRP)-coupled antibodies were incubated for 1 h at room temperature. The blots were developed using WesternBright™ enhanced chemiluminescence (ECL) (Advansta Inc., San Jose, USA; K-12045-D50) and the Fusion FX7 (Vilber Lourmat SAS, Collégien, France) image acquisition system.

For mammalian cells, 500,000 cells expressing mApple-RAB5 and GFP-RAB7 were lysed in RIPA buffer (50 mM Tris pH 8, 150 mM NaCl, 5 mM EDTA, 0.5% sodium deoxycholate, 0.1% SDS, 1% Triton X-100) supplemented with 1x Halt's protease and phosphatase inhibitor cocktail. Lysates were mixed with 4x Laemmli buffer to reach 1x final concentration of Laemmli buffer. Western blot was performed as mentioned above. All antibodies used for western blot analysis are listed in Appendix Table S16 and the Reagents and Tools Table.

## Cell culture, transfection procedure, and CRISPR-Cas9 KO in mammalian cells

HeLa CCL2 cells were grown at 37 °C and 5% CO$_2$ in high-glucose Dulbecco's modified Eagle's medium (DMEM) (Sigma-Aldrich, Merck KGaA), supplemented with 10% fetal bovine serum (FBS) (Biowest SAS, Nuaillé, France), 2 mM L-Glutamine (Carl Roth GmbH + Co. KG., Karlsruhe, Germany), 1 mM Sodium Pyruvate (Gibco, Thermo Fisher Scientific Inc.), and 1x Penicillin and Streptomycin (Sigma-Aldrich, Merck KGaA). The HeLa cell line was a kind gift of Martin Spiess and its identity was authenticated by short tandem repeat (STR) analysis from an external service provider (Microsynth AG). All cell lines created and used for the study were confirmed to be mycoplasma-negative via PCR and are listed in Appendix Table S12 and the Reagents and Tools Table.

For transient cell transfections, cells were plated into 6-well plates to reach 70% confluency the following day and transfected with 1 μg plasmid DNA complexed with Helix-IN transfection reagent (OZ Biosciences SAS, Marseille, France).

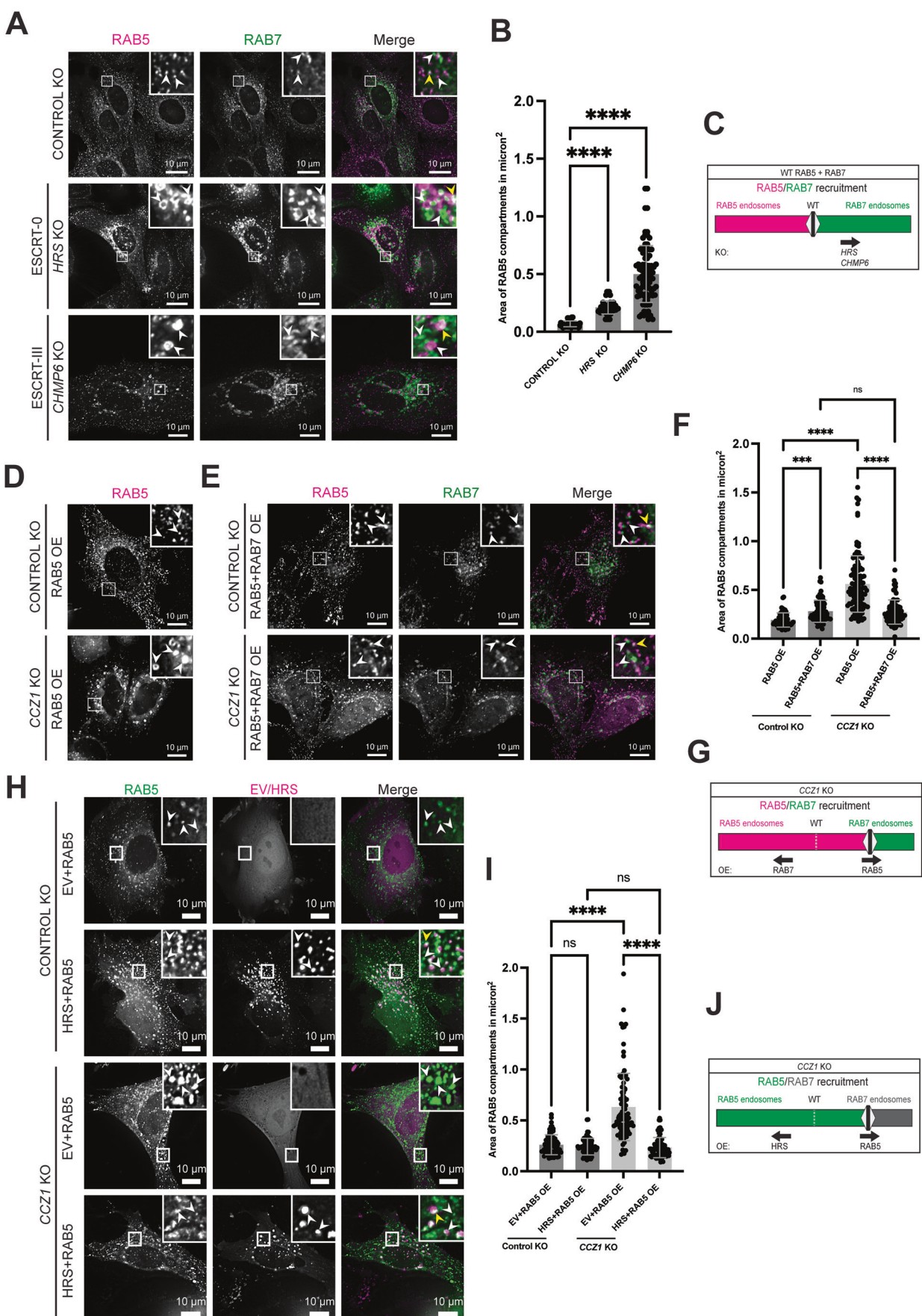

**Figure 8.   The coordination of ESCRT mediated ILV formation and Rab conversion and the second Rab7GEF downstream of MON1/CCZ1 are conserved in mammals.**

(A) Enlarged Rab5 (magenta) and Rab7 (green) vesicles in *HRS* KO and *CHMP6* KO HeLa cells stably expressing mApple-Rab5 and GFP-Rab7. White arrowheads mark Rab5 and Rab7 positive structures, respectively in the close ups of the individual channels. Yellow arrowheads mark colocalization events in the merge insets. (B) Quantification of data in (A). The size of Rab5 positive structures (magenta), shown in micron$^2$, increases in both of the knockouts compared to control knockout. Each data point represents an average area of one ROI. Obtained *P*-values: Control KO vs. *HRS* KO $P = 6.1E-6$, Control KO vs. *CHMP6* KO $P < 1.0E-15$. (C) Slider model, illustrating the effect of *HRS* or *CHMP6* knockout on the recruitment of Rab5 and Rab7 to endosomal structures (A and B). The depicted WT level corresponds to the Control KO. (D) *CCZ1* KO causes the formation of enlarged Rab5 positive structures (magenta) in HeLa cells compared to the control knockout cells. Rab5 positive structures are highlighted with white arrowheads in the insets. (E) Overexpression of GFP-Rab7 rescues the enlarged size of mApple-Rab5 positive structures in *CCZ1* KO cells. Rab5 and Rab7 positive structures, respectively, are highlighted via white arrowheads in the insets of the individual channels. In the insets of the merged images, colocalization events are marked via yellow arrowheads. (F) Quantification corresponding to (D) and (E). Reduction in the area of Rab5 positive structures (magenta) when GFP-Rab7 is overexpressed. Each data point represents an average area of one ROI. *P*-values: Rab5 OE (Control KO) vs. Rab5+Rab7 OE (Control KO) $P = 0.0003$, Rab5 OE (Control KO) vs. Rab5 OE (*CCZ1* KO) $P < 1.0E-15$, Rab5+Rab7 OE (Control KO) vs. Rab5+Rab7 OE (*CCZ1* KO) $P > 0.9999$, Rab5 OE (*CCZ1* KO) vs. Rab5+Rab7 OE (*CCZ1* KO) $P = 3.4E-13$. (G) Slider model, showing the effect of Rab5 or Rab7 overexpression on the recruitment of Rab5 and Rab7 to endosomal structures in *CCZ1* KO background (D–F). The WT level is shown as gray dotted line for comparison. (H) Overexpression of HRS-RFP rescues the enlarged size of GFP-Rab5 positive structures in *CCZ1* KO cells. In the close ups of the individual channels Rab5 and HRS, respectively, are marked via white arrowheads. Yellow arrowheads highlight colocalization events in the insets of the merged images. (I) Quantification corresponding to (H). Reduction in the area of GFP-Rab5 positive structure upon overexpression of HRS-RFP in *CCZ1* KO cells. Each data point represents an average area of one ROI. *P*-values: EV+Rab5 OE (Control KO) vs. HRS+Rab5 OE (Control KO) $P = 0.7955$, EV+Rab5 OE (Control KO) vs. EV+Rab5 OE (*CCZ1* KO) $P < 1.0E-15$, HRS+Rab5 OE (Control KO) vs. HRS+Rab5 OE (*CCZ1* KO) $P = 0.9912$, EV+Rab5 OE (*CCZ1* KO) vs. HRS+Rab5 OE (*CCZ1* KO) $P < 1.0E-15$. (J) Slider model, displaying the effect of Rab5 or HRS overexpression on the recruitment of Rab5 on endosomal structures in *CCZ1* KO background (H and I). The WT level is shown as gray dotted line for comparison. Data information: Merged images were individually adjusted in all panels. Representative pictures with magnifications (white box) on the right upper corner are shown for each condition (scale bars: 10 μm) in (A, D, E and H); $n = 3$ independent experiments. For the quantifications shown in (B, F and I) 1500 structures were measured each ($n = 1500$). Kruskal–Wallis test with Dunn's multiple comparisons test was performed for each experiment to determine the statistical significance between the examined conditions (B and F). For (I), a Brown–Forsythe ANOVA test and Welch's ANOVA test, followed by Dunnett's T3 multiple comparisons test were performed for the same purpose. Significance levels are displayed in the graphs as ***$P \leq 0.001$, ****$P \leq 0.0001$ and ns $P > 0.05$. The data show the mean ± s.d. for all examined conditions (B, F and I). Unprocessed images and statistical raw data are available as source data. Source data are available online for this figure.

For CRISPR/Cas9-mediated knockout, guide RNAs were selected using the CRISPR design tool CHOPCHOP (see Appendix Table S17 and the Reagents and Tools Table). Two guide RNAs were designed from two different exons for each target gene (*HRS*: exon 1 and exon 22; *CHMP6*: exon 3 and exon 5). Annealed oligonucleotides were cloned into two different plasmids containing mTagBFP2 marker and Puromycin resistance, respectively (Px458 mTagBFP2 and Px459 Puro). For primer sequences see Appendix Table S13. Constructs were verified by sequencing (Microsynth AG). In brief, HeLa cells stably expressing low levels of mApple-Rab5 and GFP-Rab7 (Fig. EV4H) were seeded at $1 \times 10^6$ cells per 10 cm dish. The following day, cells were transfected with 2.5 μg of the plasmids (control vectors without insert or vectors containing a guide RNA against target gene). Transfecting media was exchanged with fresh media after 6 h. Cells were treated with puromycin for 24 h after transfection followed by fluorescence-activated cell sorting (FACS) on the next day. For FACS, 48 h after transfection, cells were trypsinized and resuspended in cell-sorting medium (2% FBS and 2.5 mM EDTA in phosphate-buffered saline (PBS)) and sorted on a BD FACS AriaIII Cell Sorter (Becton, Dickinson and Company Corp., Franklin Lakes, USA). GFP, mApple and BFP positive cells were collected and seeded in a new well.

## Live cell imaging of mammalian cells

For live imaging, cells were plated in 8 well chambered coverslip, and media was replaced with warm imaging buffer (5 mM dextrose (D(+)-glucose, $H_2O$), 1 mM $CaCl_2$, 2.7 mM KCl, 0.5 mM $MgCl_2$ in PBS) just before imaging. Images were taken at 37 °C on an inverted Axio Observer Zeiss microscope (Carl Zeiss AG) equipped with a TempModule S, a Heating Unit XL S, an incubator XLmulti S1 (PeCon GmbH, Erbach, Germany), a HXP 120 V (Leistungse-lektronik JENA GmbH, Jena, Germany) and a SMC 2009 (Carl

Zeiss AG). For imaging a Plan Apochromat N 63×/1.40 oil DIC M27 objective with a Photometrics Prime 95B camera (Teledyne Photometrics Inc., Tucson, USA) was used. Z-stack images were captured using the Zen Blue 2.6 imaging software (Carl Zeiss AG) and deconvolved using Huygens Remote Manager (HRM) (Huygens Software (Scientific Volume Imaging B.V., Hilversum, Netherlands)). Acquired images were processed by using the OMERO client server web tool and Fiji.

## Immunostaining

To visualize the effect of Rabex5 WT and mutants on Rab5 structures in control KO and *CCZ1* KO HeLa, cells were plated onto a coverslip 24 h prior to transfection. Transfection was performed with myc-tagged plasmids of Rabex5 or Rabex5 mutants together with mApple-Rab5, which only led to very small increase in Rab5 expression (Fig. EV4I). 24 h post transfection, cells were fixed with 4% paraformaldehyde and blocked with 5% FBS in PBS. Coverslips were stained with anti c-myc antibody (1:200) in 5% FBS in PBS for 1 h. Prior to the staining with secondary antibody, coverslips were rinsed with PBS. Anti-mouse conjugated with Alexa Fluor™ 488 (1:500) was used in 5% FBS in PBS for 1 h. Coverslips were rinsed 3 times with PBS, mounted on slides with Fluoromount-G (Southern Biotechnology Associates Inc., Birmingham, USA; 0100-01) and left to dry for 24 h before imaging. All antibodies used for immunostaining are listed in Appendix Table S16 and the Reagents and Tools Table.

## Rab5 structure size quantification in mammalian cells

Area analysis for Rab5 positive structures was done using "analyze particle" function of Fiji. Initially, Rab5 and Rab7 channels were separated using split channels function of Fiji. Z-stacks were

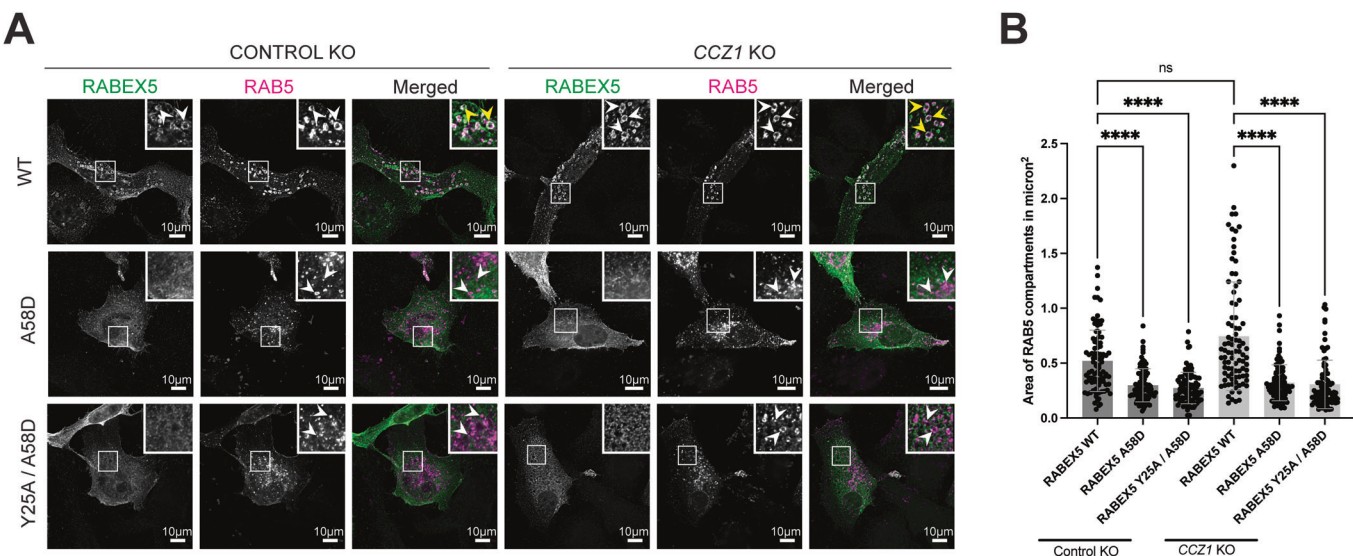

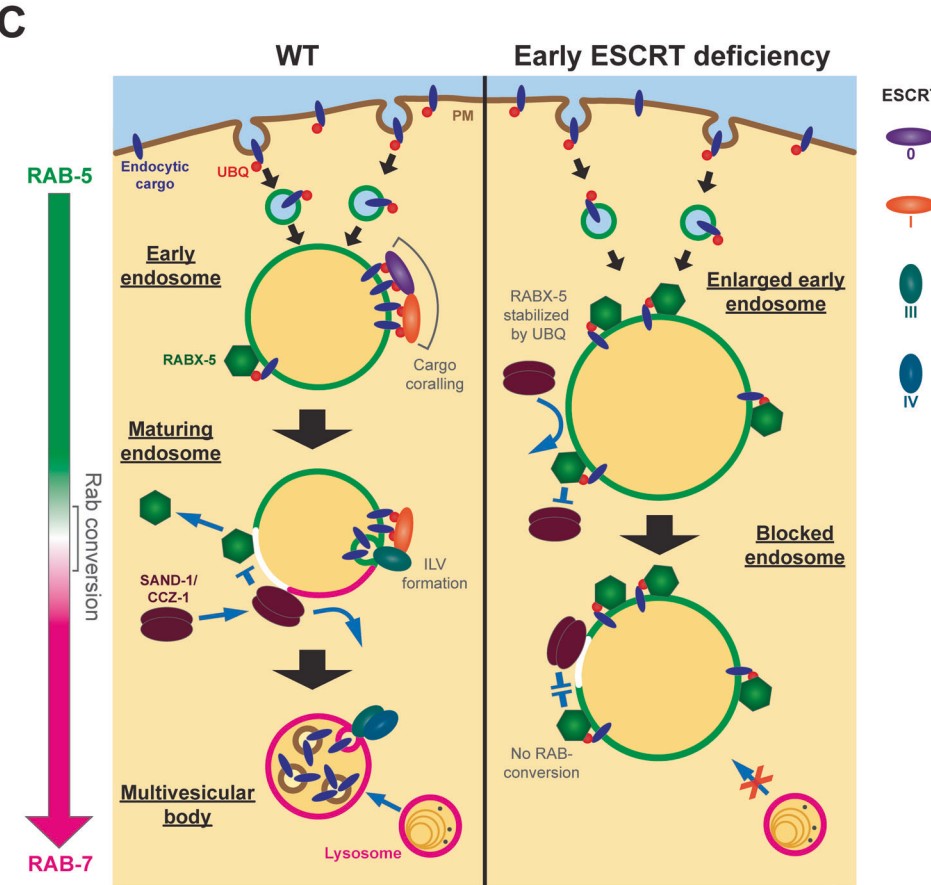

converted to a maximum projection. A ROI of 100 μm² was drawn manually and duplicated. Objects were manually selected using threshold based on the intensity. Areas of individual Rab5 positive structures in the ROI were acquired using the "analyze particle" function. The average area of Rab5 positive particles in individual ROIs was plotted using Prism (GraphPad Software LLC.).

## Plasmid construction and source for experiments and knockouts in mammalian cells

All constructs and primers used in this study are listed in the Appendix Tables S13 and S14 and the Reagents and Tools Table.

◀ **Figure 9.  Rabex5 recruitment to endosomes depends on ubiquitin binding in mammalian cells.**

(**A**) Enlarged Rab5 (magenta) early endosomes become smaller when Rabex5 carries mutations in its ubiquitin binding site. Mutated Rabex5 (green) is predominantly cytoplasmic. Similar effects can be seen in a *CCZ1* KO background where Rab conversion is blocked, showing an early upstream involvement of Rabex5 in the pathway. Some of the individual signals are marked by white arrowheads. In the merged images, colocalization events are marked by yellow arrowheads. (**B**) Quantification corresponding to data shown in (**A**). The size of Rab5 positive structures (magenta), shown in $micron^2$, decreases in the cells expressing mutated Rabex5, independent of the cell background. Each data point represents the average size of Rab5 particles in one ROI. *P*-Values: Rabex5 WT vs. Rabex5 A58D (Control KO) $P = 1.9E-7$, Rabex5 WT vs. Rabex5 Y25A / A58D (Control KO) $P = 7.2E-10$, Rabex5 WT vs. Rabex5 A58D (*CCZ1* KO) $P = 1.9E-5$, Rabex5 WT vs. Rabex5 Y25A/A58D (*CCZ1* KO) $P = 2.6E-9$, Rabex5 WT (Control KO) vs. Rabex5 WT (*CCZ1* KO) $P = 0.4052$. (**C**) Working model for the function of RABX-5 binding to ubiquitinated cargo and the resulting regulation of endosomal traffic. ESCRT function and Rab conversion are coordinated by the hand-off of cargo from RABX-5 to early ESCRT-0 and ESCRT-I complexes that corral the cargo for later ILV formation. Since the stable binding of RABX-5 keeps the endosome in its early stage with RAB-5, the presence of ubiquitinated cargo inhibits the maturation of the early endosome. Only when sufficient cargo has been funneled into the ESCRT pathway will SAND-1/CCZ-1 be able to displace RABX-5 from the membrane. This interrupts the activation of RAB-5 via RABX-5 and allows the Rab switch. A deficiency of early ESCRT will lead to a situation where RABX-5 is permanently bound to early endosomes and Rab conversion is blocked, resulting in enlarged endosomes full of ubiquitinated cargo. Data information: Merged images were individually adjusted in all panels. Representative pictures with magnifications (white box) on the right upper corner are shown for each condition (scale bars: 10 μm) in (**A**). $n = 3$ independent experiments. For the quantifications shown in (**B**), 3 ROIs were measured in each cell and 30 cells per condition were measured for the experiment. Kruskal–Wallis test followed by Dunn's multiple comparisons test was performed for each experiment to determine the statistical significance between the examined conditions. Significance levels are displayed in the graphs as ****$P \leq 0.0001$ and ns $P > 0.05$. The data show the mean ± s.d. for all examined conditions (**B**). Unprocessed images and statistical raw data are available as source data. Source data are available online for this figure.

## Plasmid construction for *C. elegans* experiments

To generate the RNAi constructs for *tsg-101*, *vps-2*, *vps-60*, and *rabx-5*, primers with homology overhangs for pDT7 (L4440) (Timmons et al, 2001; Timmons and Fire, 1998) were designed using the NEBuilder Assembly Tool (New England Biolabs Inc., Ipswich, USA) (see Appendix Table S15). Sequences were amplified by PCR from *C. elegans* cDNA or genomic DNA and cloned into the EcoRV site of pDT7 using Gibson assembly. Plasmids were verified by sequencing (Microsynth AG).

## Reagents and tools used for *C. elegans* and cell culture experiments

Important chemicals, reagents, enzymes and instruments used in this study are listed in the Reagents and Tools Table.

### Statistical analysis

Statistical analyses were performed using Prism (GraphPad Software LLC.). Used statistical tests and the determined *P*-values (in case of $P = 0.0001$ with four decimal places, two extra decimal places are given to clarify the value; in case of $P < 0.0001$, *P*-values are given as X.XE-X) are indicated in the figure legends. The data sets were tested for normality/lognormality with the Anderson-Darling test, the Shapiro–Wilk test, the Kolmogorov–Smirnov test and the D'Agostino & Pearson test, for differences in the s.d. with the Brown–Forsythe test and the Bartlett's test and for differences in the variance with the F test. For each test, a significance level (alpha) of 0.05 was used. Based on the examined statistical distribution and variance, we have chosen the appropriate statistical test, e.g., if the data were not normal distributed and multiple comparisons were required, we used a nonparametric statistical test (e.g., Kruskal–Wallis test with Dunn's multiple comparisons test). All programs used in this study are listed in the appendix (Appendix Table S17) and the Reagents and Tools Table.

## Data availability

The electron microscopy (EM) images generated in this study have been deposited in the BioImage Resource and are accessible at S-BIAD1503. All other data supporting the findings of this study are included within the manuscript.

The source data of this paper are collected in the following database record: biostudies:S-SCDT-10_1038-S44318-025-00367-7.

## Peer review information

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

## Acknowledgements

We thank Barth D. Grant, Anbing Shi, Emily R. Troemel, Attila Stetak, Juan Bonifacino and Martin Spiess for worm strains, plasmids and HeLa cells, respectively. We are grateful to Christian Ungermann and Lars Langemeyer for useful discussions and personal communications. Some strains were provided by the CGC, which is funded by the NIH Office for Research Infrastructure Programs (P40 OG010440). We acknowledge Dora Stetak for technical support and Julia K. Nussbaum and Arian Dill for their help in plasmid construction. The imaging core facility of the Biozentrum (IMCF) is acknowledged for superb support. We thank Carmen Kaiser from the Center for Microscopy and Image Analysis, University of Zurich for ultra-thin sectioning of embedded *C. elegans* for TEM. Cells were sorted in the Biozentrum FACS core facility. This work was supported by the University of Basel and the Swiss National Science Foundation (310030_197779, 310030_185127).

## Author contributions

**Daniel P Ott**: Data curation; Formal analysis; Validation; Investigation; Visualization; Methodology; Writing—original draft; Writing—review and editing. **Samit Desai**: Data curation; Formal analysis; Validation; Investigation; Visualization; Methodology; Writing—review and editing. **Jachen A Solinger**: Conceptualization; Data curation; Formal analysis; Validation; Investigation; Visualization. **Andres Kaech**: Data curation; Validation; Investigation; Methodology. **Anne Spang**: Conceptualization; Supervision; Funding acquisition; Writing—original draft; Project administration.

Source data underlying figure panels in this paper may have individual authorship assigned. Where available, figure panel/source data authorship is listed in the following database record: biostudies:S-SCDT-10_1038-S44318-025-00367-7.

## Disclosure and competing interests statement

The authors declare no competing interests.

# Expanded View Figures

**Figure EV1.   Knockdowns of core early and late ESCRT factors as well as knockdowns of additional ESCRT-III factors affect endosome maturation in WT and** *sand-1(KO)*, **related to Fig. 1.**

(A) Additional ESCRT knockdowns performed in the ESCRT RNAi screen shown in Fig. 1A. White arrowheads pointing to GFP::RAB-5 and mCherry::RAB-7 positive structures, respectively in the individual channels. In the merges colocalization events are marked via yellow arrowheads (signals: RAB-5 > green and RAB-7 > magenta). (B) Further ESCRT knockdowns performed in the ESCRT RNAi screen shown in Fig. 1B. Consistent with data shown in Fig. 1B, knockdowns of further core early ESCRTs or a core late ESCRT factor cause no further enlargement of the through *sand-1(KO)* enlarged GFP::RAB-5 positive structures and have only minor effects on the colocalization of GFP::RAB-5 and mCherry::RAB-7. Similar effects on the colocalization of GFP::RAB-5 and mCherry::RAB-7 and the GFP::RAB-5 structure size are also observable in knockdowns of additional ESCRT-III factors. White arrowheads pointing to GFP::RAB-5 and mCherry::RAB-7 positive structures, respectively in the individual channels. In the merges colocalization events are marked via yellow arrowheads (signals: RAB-5 > green and RAB-7 > magenta). Data information: Merges were individually adjusted in all panels. Representative pictures with magnifications (white box) on the right are shown for each experiment (scale bars: 10 µm (main pictures) and 1 µm (magnifications)). Unprocessed images are available as source data. Source data are available online for this figure.

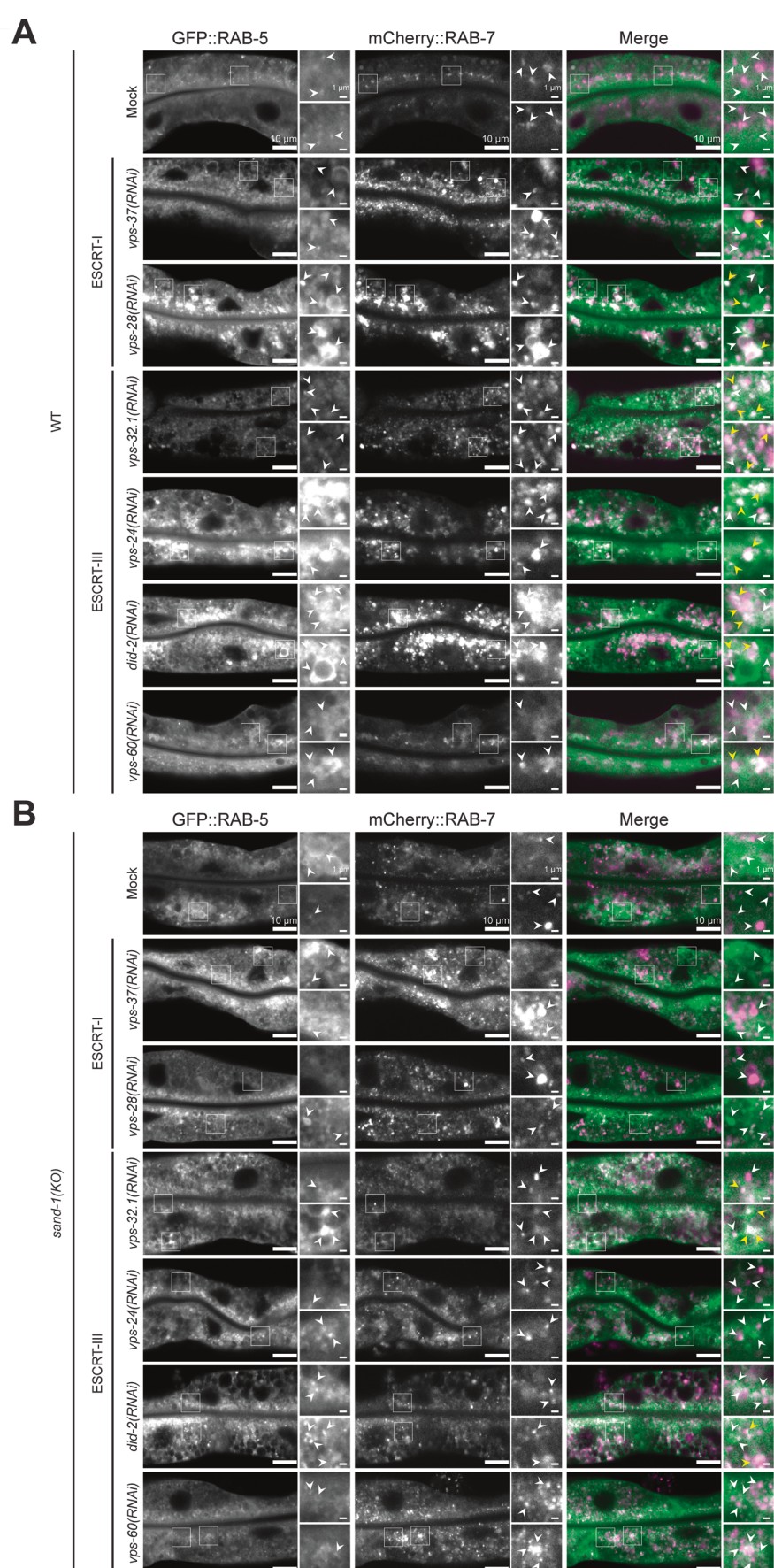

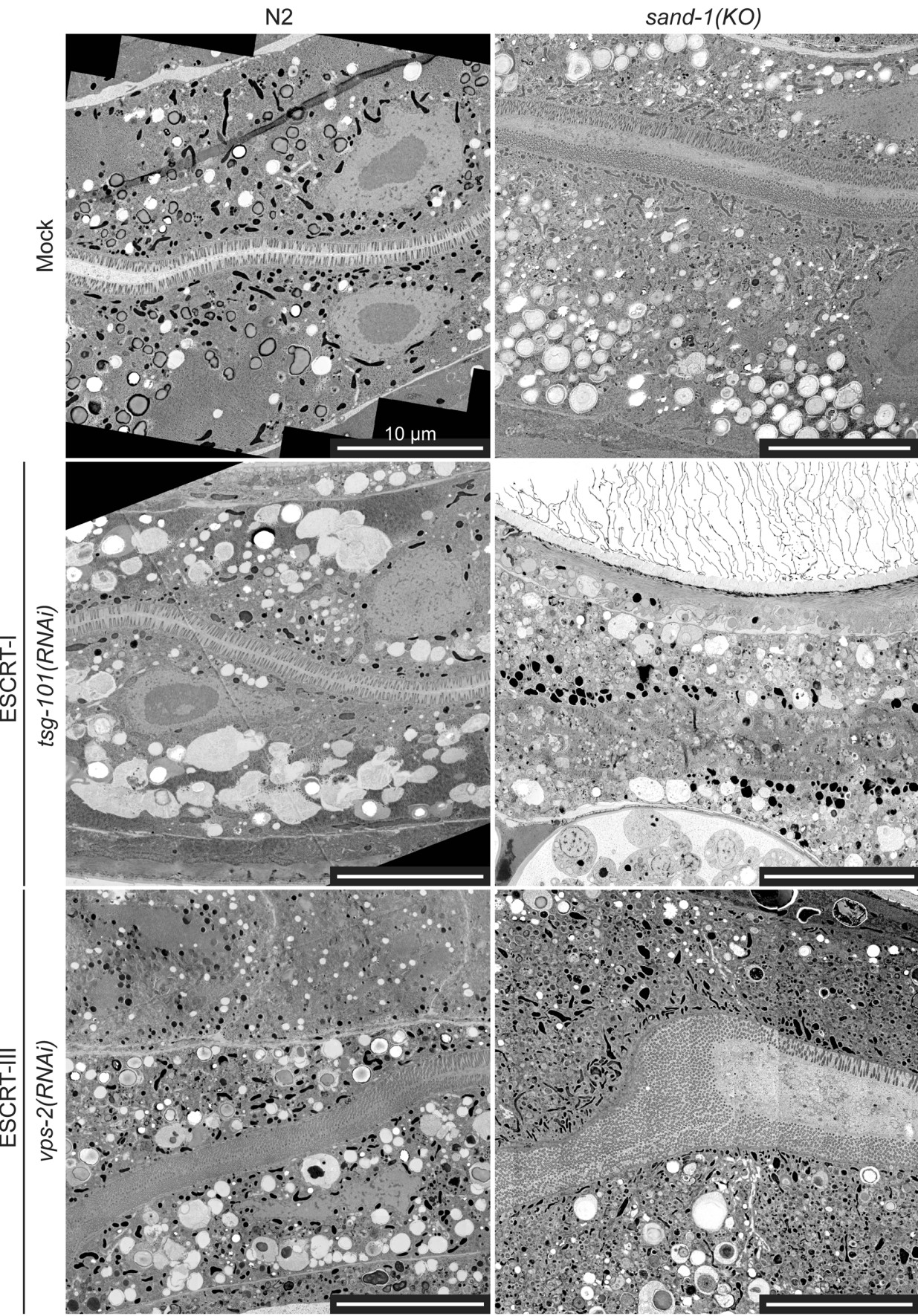

◄ **Figure EV2.   ESCRT knockdowns affect the intracellular morphology in WT and *sand-1(KO)*, related to Fig. 2.**

TEM overview pictures belonging to experiments shown in (Fig. 2A, C, E). The *sand-1(KO)* causes the formation of large granular structures accumulating in the basal region but does not increase the MVB size. Knockdown of *tsg-101* increases the size of endosomal structures, whereas *vps-2* knockdown causes no major changes in this regard but leads to the formation of additional small and big granular structures. The observed ESCRT RNAi effect are strain background independent. Data information: Representative overview TEM pictures are shown for each examined condition. Overview electron micrographs are individually adjusted and were generated with unstitched data sets (scale bars 10 μm). Unprocessed images are available as source data and online in the BioImage Resource.

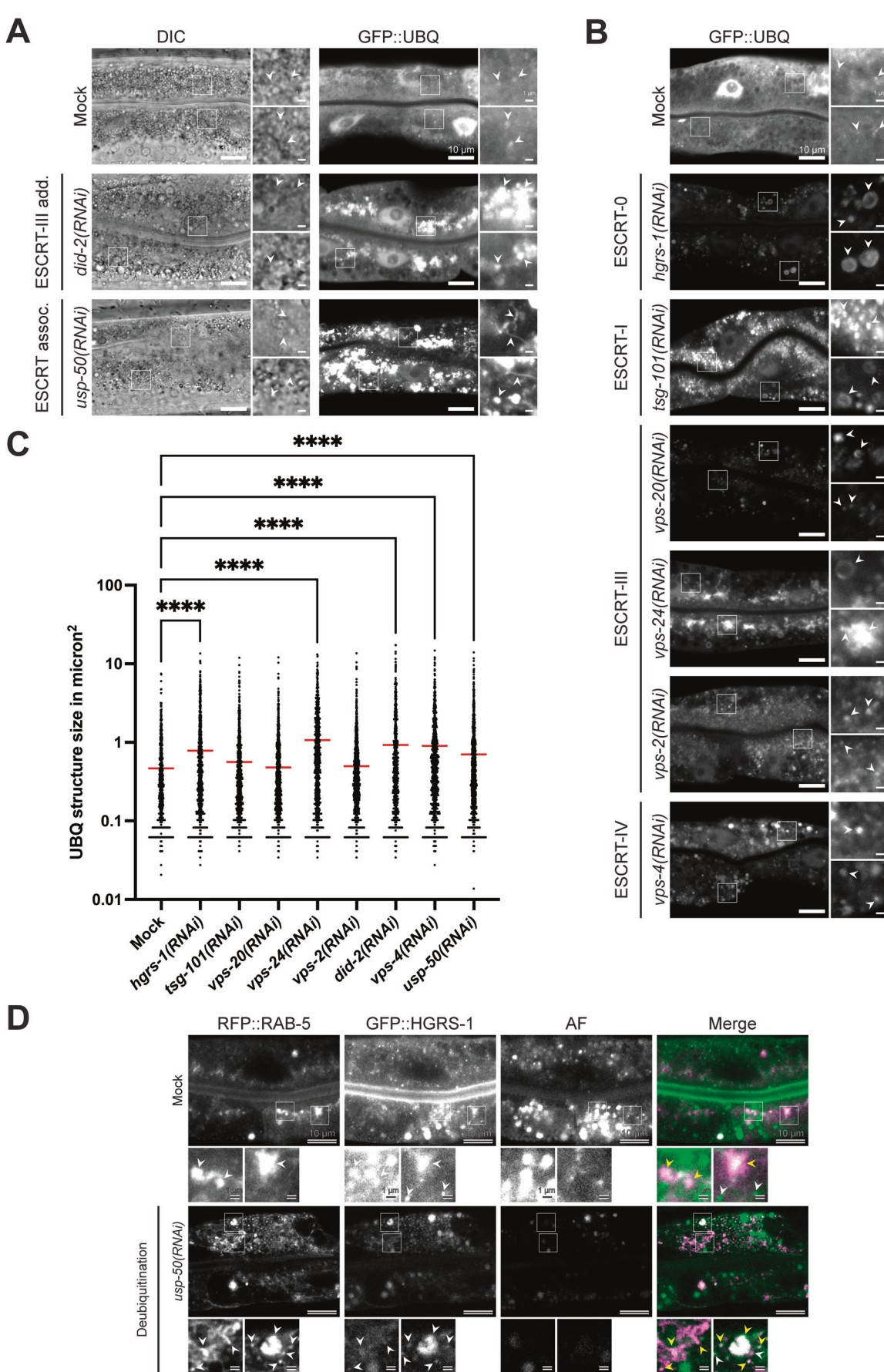

◀ **Figure EV3.** *usp-50* and *did-2* knockdown cause UBQ aggregation in the cytoplasm and USP-50 presence is required for HGRS-1 stability in *C. elegans*, related to Fig. 4.

(A) The knockdown of *did-2* has only minor effects on nuclear GFP::UBQ levels and cause the formation of GFP::UBQ accumulations in addition to enlarged structures. *usp-50(RNAi)* however, generates GFP::UBQ accumulations, enlarged structures, tubular networks and strongly reduces nuclear GFP::UBQ levels. White arrowheads pointing to GFP::UBQ positive structures and corresponding areas in the DIC, respectively. (B) Images of early and late ESCRT knockdowns from Fig. 4 adjusted to equal brightness settings. *hgrs-1(RNAi)* and *vps-20(RNAi)* show strongly reduced GFP::UBQ levels in comparison to Mock as well as to the other ESCRT knockdowns. Some of the individual signals are marked by white arrowheads. (C) Quantification of the GFP::UBQ positive structure sizes belonging to (A) and Fig. 4A. The individual sizes of the measured GFP::UBQ positive structures are displayed for each examined condition in micron$^2$. For analysis shown in the graph 10 worms per condition ($n = 10$) and more than 9500 structures in total ($n > 9500$) were examined ($n = 3$ independent experiments). The mean particle size is shown for each condition as red line in the graph and the data are shown in log 10 scale (Y-axis). Obtained *P*-values: Mock vs. *hgrs-1(RNAi)* $P = 2.2E{-}6$, Mock vs. *tsg-101(RNAi)* $P = 0.3887$, Mock vs. *vps-20(RNAi)* $P > 0.9999$, Mock vs. *vps-24(RNAi)* $P < 1.0E{-}15$, Mock vs. *vps-2(RNAi)* $P = 0.1113$, Mock vs. *did-2(RNAi)* $P = 6.2E{-}9$, Mock vs. *vps-4(RNAi)* $P < 1.0E{-}15$, Mock vs. *usp-50(RNAi)* $P = 4.5E{-}7$. (D) The knockdown of *usp-50* causes a strong reduction of the GFP::HGRS-1 signal and generates tubular, network like RFP::RAB-5 positive structures and RFP::RAB-5 positive aggregates. In addition, some of the remaining GFP::HGRS-1 positive structures are also positive for RFP::RAB-5. White arrowheads marking GFP::HGRS-1 and RFP::RAB-5 positive structures, respectively in the individual channels. Yellow arrowheads indicating colocalization events in the merges (Signals: HGRS-1 > green and RAB-5 > magenta). The autofluorescence channel (AF) is not shown in the merge for simplification. Data information: Representative pictures with corresponding enlargements (white box) next to it are shown for each experiment (scale bars: 10 μm (main pictures) and 1 μm (enlargements)) in (A, B and D). Belonging DIC pictures elucidate the extent of the gut, marker independent are depicted for all experiments showing in (A). Kruskal–Wallis test with Dunn's multiple comparisons test was performed to determine the statistical significance between the examined conditions (C). Significance levels are displayed in the graph as ****$P \leq 0.0001$. Comparisons with $P > 0.05$ are not shown for simplification. Unprocessed images and statistical raw data are available as source data. Source data are available online for this figure.

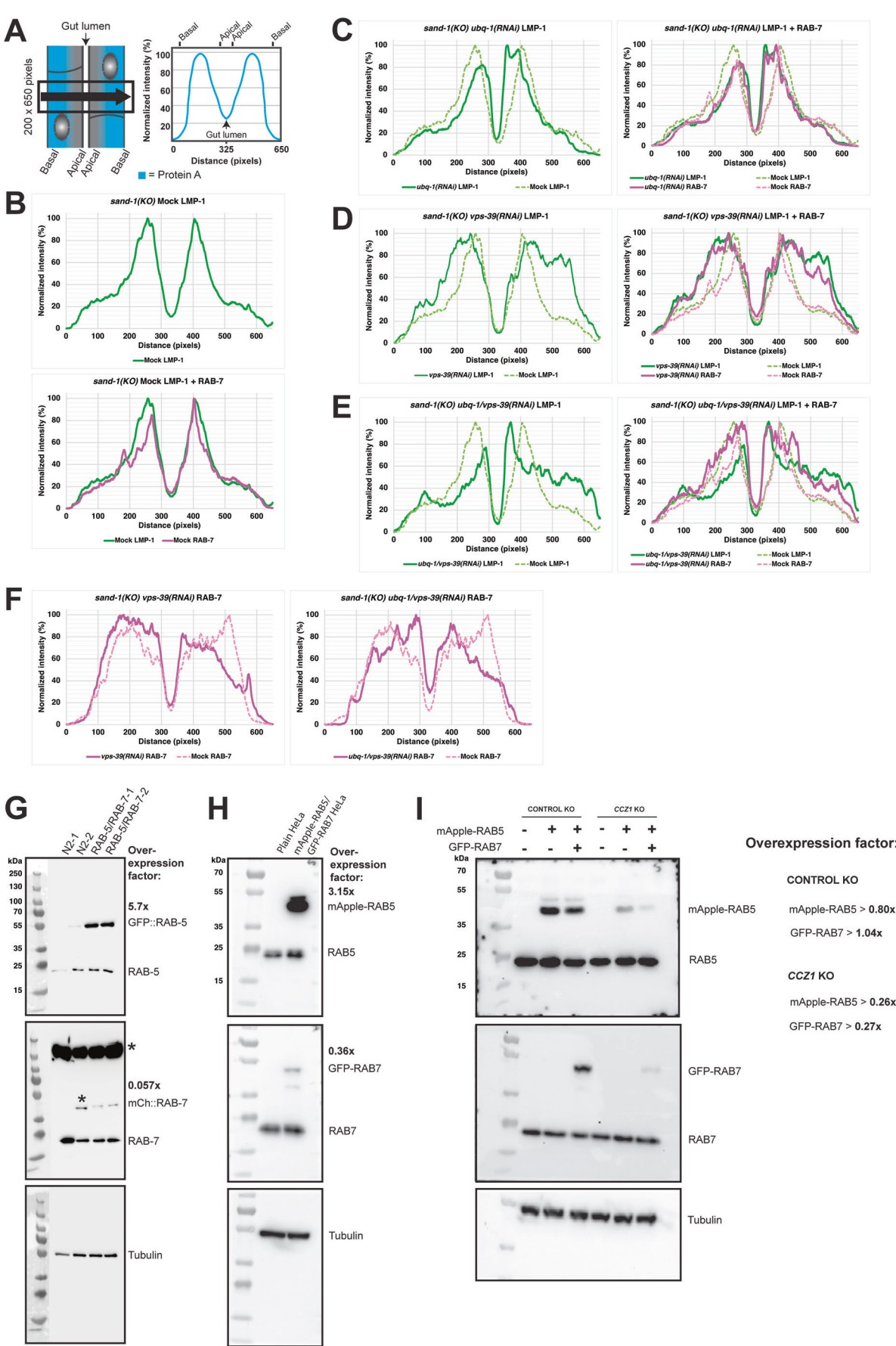

◄ **Figure EV4. LMP-1 and RAB-7 area localizations in *sand-1(KO)* under reduced UBQ levels and knockdowns of *vps-39* and western blots of tagged RAB proteins in *C. elegans* strains and cell lines, related to Figs. 5, 6 and worm and cell culture experiments.**

(A) Schematic representation of the approach used to generate the area plots shown in Figs. 5, 6 and EV4. A simplified worm gut with intestinal cells expressing a blue example protein is shown on the left side. The localization of this protein gets measured via a ROI (200 × 650 pixels) spanning the whole worm gut. This creates an area plot, highlighting the localization of the protein in the intestinal cells, shown on the right side. (B, C) Area plots belonging to Fig. 5I, K. Overexpression of mCherry::RAB-7 causes an apical localization of LMP-1::GFP which can be further augmented through RNAi mediated UBQ level reduction. (D, E) Area plots belonging to Fig. 6C, D. The knockdown of HOPS subunit *vps-39* abolishes the partial rescue of *sand-1(KO)* mediated by the mCherry::RAB-7 overexpression and causes a more basal distribution of LMP-1::GFP. This localization shift is less prominent if the UBQ levels are reduced together with the *vps-39* knockdown. (F) Examined mCherry::RAB-7 area localization belonging to Fig. 6A. The knockdown of *vps-39* impairs the recruitment of mCherry::RAB-7 to apical structures caused by the reduction of UBQ levels. The control group is the same like shown in Fig. 5B and the data were collected and analyzed like described for this experiment. (G) Western blot analysis of overexpression in worms carrying GFP::RAB-5 and mCherry::RAB-7 inserted transgenes (used in Fig. 1, Fig. 5, Fig. 6, Fig. 7, Fig. EV1 and Fig. EV5). The indicated worm lysates were loaded ($n = 2$ biological replicates) for N2 (wild-type) and RAB-5/RAB-7 worms with transgenes. The relative overexpression was calculated by the ratio between the endogenous lower band and the tagged upper band. The overexpression factor is indicated and is the mean of the two replicates (GFP::RAB-5: 5.7x, mCherry::RAB-7: 0.057x). The asterisks denote unspecific bands in the RAB-7 blot (middle panel). Tubulin was used as a loading control. (H) Western blot analysis of overexpression in HeLa cells stably expressing mApple-RAB5 and GFP-RAB7 was done (used in Fig. 8). The cell lysates from control and stably expressing mApple-RAB5 and GFP-RAB7 were separated by SDS PAGE. The relative overexpression was calculated by the ratio between the endogenous lower band and the tagged upper band. The overexpression factor is indicated and is a mean of three replicates (mApple-RAB5: 3.15x, GFP-RAB7: 0.36x). Tubulin was used as a loading control. (I) Western blot analysis of overexpression of transiently transfected mApple-RAB5 and GFP-RAB7 in Control KO and CCZ1 KO cells was done (used in Fig. 8). The cell lysates from aforementioned conditions were separated by SDS PAGE. The relative overexpression was calculated by the ratio between the endogenous lower band and the tagged upper band. The overexpression factor is indicated and is a mean of three replicates (Control KO > mApple-RAB5: 0.8x, GFP-RAB7: 1.04x; *CCZ1* KO > mApple-RAB5: 0.26x, GFP-RAB7: 0.27x). Tubulin was used as a loading control. Data information: The individual mCherry::RAB-7 area plots for the merge area plots (B–E) are shown in Figs. 5K and 6D. Statistical raw data are available as source data (B–I). Source data are available online for this figure.

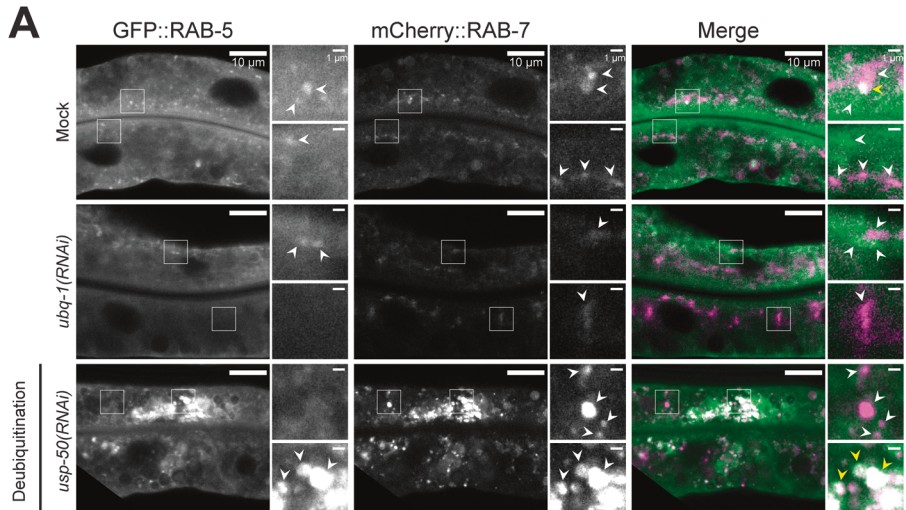

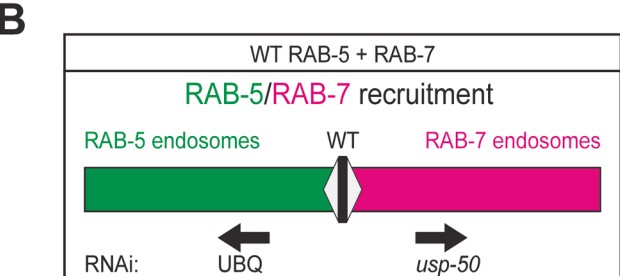

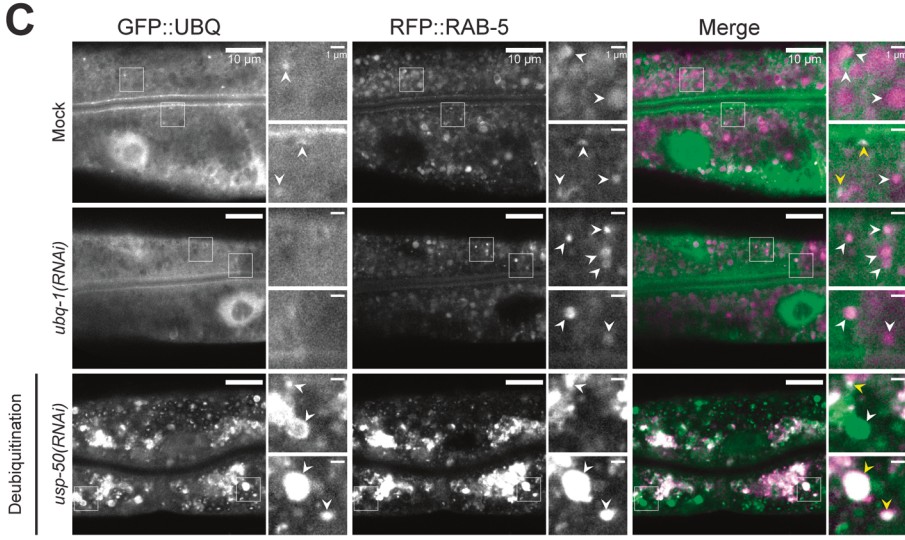

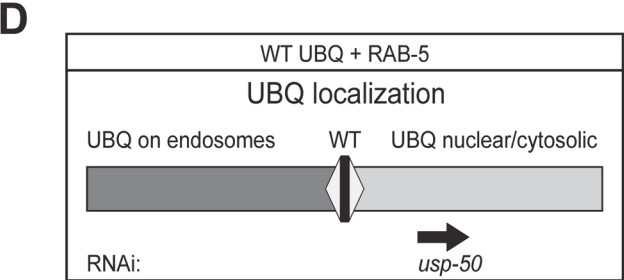

◀ **Figure EV5. UBQ reduction reduces the number of RAB-5 structures and *usp-50(RNAi)* causes colocalization of RAB-5 with RAB-7 and UBQ.**

(A) The reduction of the UBQ level via RNAi causes a depletion of GFP::RAB-5 positive structures. Knockdown of *usp-50* causes a mislocalization of GFP::RAB and mCherry::RAB-7 and increases their colocalization. White arrowheads marking GFP::RAB-5 and mCherry::RAB-7 positive structures, respectively in the individual channels. Yellow arrowheads marking colocalization events in the merges (Signals: RAB-5 > green and RAB-7 > magenta). (B) Schematic representation of the endosomal RAB-5/ RAB-7 balance in a *C. elegans* intestinal cell in WT background. Effects of UBQ reduction or *usp-50(RNAi)* on this balance are shown via black arrows. (C) The RNAi mediated UBQ level abatement causes the formation of smaller RFP::RAB-5 aggregates and leads to less prominent GFP::UBQ structures. *usp-50(RNAi)* causes the formation of large GFP::UBQ structures and aggregates which are often also RFP::RAB-5 positive and reduces the GFP::UBQ signal in the nucleus. White arrowheads marking GFP::UBQ and RFP::RAB-5 positive structures, respectively in the individual channels. Yellow arrowheads marking colocalization events in the merges (Signals: UBQ > green and RAB-5 > magenta). (D) Schematic representation of the UBQ distribution in an intestinal *C. elegans* cell expressing marked UBQ and RAB-5 in WT background. The effect of the *usp-50* knockdown on this equilibrium is shown via a black arrow. Data information: Merges were individually adjusted in all panels. Representative pictures with close ups (white box) on the right are shown for each experiment (scale bars: 10 μm (main pictures) and 1 μm (close ups) in (A and C); $n = 3$ independent experiments. Unprocessed images are available as source data. Source data are available online for this figure.

