## [Peer Review File · The EMBO Journal]

Coordination between ESCRT function and Rab conversion during endosome maturation

Daniel Ott, Samit Desai, Jachen Solinger, Andres Kaech, and Anne Spang

Corresponding author(s): Anne Spang (anne.spang@unibas.ch)

Review Timeline:

Submission Date:	30th Apr 24
Editorial Decision:	14th Jun 24
Revision Received:	31st Oct 24
Editorial Decision:	2nd Dec 24
Revision Received:	12th Dec 24
Accepted:	2nd Jan 25

Editor: William Teale

Transaction Report:

Dear Anne,

Thank you for submitting your manuscript entitled 'Coordination between ESCRT function and Rab conversion during endosome maturation' (EMBOJ-2024-117745) for consideration by the EMBO Journal. We have now received the reports from the referees, which I copy below.

All referees agree that the work is based on a technically accomplished and well-described collection of experiments. However, the feedback was not unambiguously positive. Issues were identified with key aspects of the work, including the proposed GEF function of Vps39, and the ability of your data to differentiate between hypotheses on how the RABX-5-lysosome interaction could be being regulated by ESCRT complexes.

I think it would be best if we discussed the reports by Zoom at the end of next week, after you have had time to fully digest the reports, and explored the most appropriate way forward. In case we decide that the key concerns of referees 2 and 3 can be addressed within a realistic time-frame, I include the instructions for revisions below.

I would also like to point out that as a matter of policy, competing manuscripts published during this period will not be taken into consideration in our assessment of the novelty presented by your study ("scooping" protection). We have extended this 'scooping protection policy' beyond the usual 3 month revision timeline to cover the period required for a full revision to address the essential experimental issues. Please contact me if you see a paper with related content published elsewhere to discuss the appropriate course of action.

When preparing any letter of response to the referees' comments, please bear in mind that this will form part of the Review Process File, and will therefore be available online to the community. For more details on our Transparent Editorial Process, please visit our website: <https://www.embopress.org/page/journal/14602075/authorguide#transparentprocess>

Best regards,

William

William Teale, Ph.D.
Editor
The EMBO Journal

When submitting your revised manuscript, please carefully review the instructions below and include the following items:

- 1) a .docx formatted version of the manuscript text (including legends for main figures, EV figures and tables). Please make sure that the changes are highlighted to be clearly visible.
- 2) individual production quality figure files as .eps, .tif, .jpg (one file per figure).
- 3) a .docx formatted letter INCLUDING the reviewers' reports and your detailed point-by-point response to their comments. As part of the EMBO Press transparent editorial process, the point-by-point response is part of the Review Process File (RPF), which will be published alongside your paper.
- 4) a complete author checklist, which you can download from our author guidelines ([https://wol-prod-cdn.literatumonline.com/pb-assets/embo-site/Author Checklist%20-%20EMBO%20J-1561436015657.xlsx](https://wol-prod-cdn.literatumonline.com/pb-assets/embo-site/Author%20Checklist%20-%20EMBO%20J-1561436015657.xlsx)). Please insert information in the checklist that is also reflected in the manuscript. The completed author checklist will also be part of the RPF.
- 5) Please note that all corresponding authors are required to supply an ORCID ID for their name upon submission of a revised manuscript.
- 6) We require a 'Data Availability' section after the Materials and Methods. Before submitting your revision, primary datasets produced in this study need to be deposited in an appropriate public database, and the accession numbers and database listed under 'Data Availability'. Please remember to provide a reviewer password if the datasets are not yet public (see <https://www.embopress.org/page/journal/14602075/authorguide#datadeposition>). If no data deposition in external databases is needed for this paper, please then state in this section: This study includes no data deposited in external repositories. Note that the Data Availability Section is restricted to new primary data that are part of this study.

Note - All links should resolve to a page where the data can be accessed.

8) For data quantification: please specify the name of the statistical test used to generate error bars and P values, the number (n) of independent experiments (specify technical or biological replicates) underlying each data point and the test used to calculate p-values in each figure legend. The figure legends should contain a basic description of n, P and the test applied. Graphs must include a description of the bars and the error bars (s.d., s.e.m.).

9) We would also encourage you to include the source data for figure panels that show essential data. Numerical data can be provided as individual .xls or .csv files (including a tab describing the data). For 'blots' or microscopy, uncropped images should be submitted (using a zip archive or a single pdf per main figure if multiple images need to be supplied for one panel). Additional information on source data and instruction on how to label the files are available at .

10) We replaced Supplementary Information with Expanded View (EV) Figures and Tables that are collapsible/expandable online (see examples in <https://www.embopress.org/doi/10.15252/embj.201695874>). A maximum of 5 EV Figures can be typeset. EV Figures should be cited as 'Figure EV1, Figure EV2' etc. in the text and their respective legends should be included in the main text after the legends of regular figures.

12) Our journal encourages inclusion of *data citations in the reference list* to directly cite datasets that were re-used and obtained from public databases. Data citations in the article text are distinct from normal bibliographical citations and should directly link to the database records from which the data can be accessed. In the main text, data citations are formatted as follows: "Data ref: Smith et al, 2001" or "Data ref: NCBI Sequence Read Archive PRJNA342805, 2017". In the Reference list, data citations must be labeled with "[DATASET]". A data reference must provide the database name, accession number/identifiers and a resolvable link to the landing page from which the data can be accessed at the end of the reference. Further instructions are available at .

We realize that it is difficult to revise to a specific deadline. In the interest of protecting the conceptual advance provided by the work, we recommend a revision within 3 months (12th Sep 2024). Please discuss the revision progress ahead of this time with the editor if you require more time to complete the revisions. Use the link below to submit your revision:

Link Unavailable

Referee #1:

EMBOJ-2024-117745

Mechanisms of trafficking through endosomes are intensely studied, which has given rise to a controversy between endosomal maturation versus vesicular trafficking. This has stirred investigations into molecular mechanisms as a way forward into formulating and testing precise predictions from both models. In the current manuscript, Spang and colleagues have addressed the question of how the staging between intraluminal vesicle formation and conversion from Rab5 to Rab7 endosomes could be achieved molecularly. Based on the evidence that is presented in the current manuscript, it appears that the Rab5 GEF Rabex5 and the Rab7 GEF Mon1/Ccz1/SAND1 are a key components for this staging. A model arises according to which corralling of ubiquitinated cargoes on maturing sorting endosomes liberates the ubiquitin-binding Rabex5 such that it can now be displaced by Mon1/Ccz1/SAND1 to operate Rab conversion.

This manuscript appears exceptionally mature. It addresses a timely problem from different angles, in different experimental systems and with a wealth of techniques by combining cell biological evidence from immunofluorescence and electron microscopy experiments with genetic testing. The insights from this substantial body of work into endosomal maturation are of interest to a general molecular and cellular biology readership. The manuscript is well written and pleasant to read. It provides a new model on a molecular circuit for a key step in endosomal maturation that likely operates in many organisms.

Referee #2:

The manuscript of Ott et al. focuses on the spatial coordination between ESCRT function and Rab conversion during the endosomal maturation process. The authors show that loss of early ESCRT subunits leads to large Rab5-positive endosomes which do not undergo Rab conversion to Rab7 positive structures. This phenotype was alleviated by reduction of ubiquitinated cargo on endosomes. The authors further show that displacement of the Rab5 GEF RABX-5 by the Rab7 GEF SAND-1/CCZ-1 does not take place in the presence of ubiquitinated cargo. Additionally, they provide data that overexpression of Rab7 partially rescues this phenotype and suggest that the HOPS complex might as Rab7 GEF in this scenario. The authors therefore reveal a hierarchy of ESCRT recruitment and Rab conversion.

This is a thorough study that further analyzes the correlation between ILV formation and Rab conversion on endosomes to define the spatial regulation of endosomal maturation. Key data are obtained using fluorescence and transmission electron microscopy and precisely quantified. The main observations in *C. elegans* are confirmed in a mammalian cell line. Overall, this manuscript provides new insights into the special hierarchy of ILV formation and Rab conversion. However, given the methodology it is a rather descriptive study, and some of the findings reported in this manuscript are confirmatory, including the reduced ILV content and enlarged size of ESCRT-depleted endosomes, and the accumulation of ubiquitinated proteins on ESCRT-depleted endosomes. Further, the identification of VPS39 as a novel GEF directly activating Rab7 is rather speculative

based on the performed microscopy experiments. The idea that Rab conversion is coordinated with ESCRT function via Rabex5 is not novel, although the proposed mechanism, Rabex5 recruitment via endosomal accumulation of ubiquitinated cargos, is novel. However, this is a mechanism that could already be hypothesized from existing data, and the present manuscript does not provide any direct proof for it.

Major points:

1. The issue of ESCRT function vs endosome maturation has been addressed previously (Parkinson et al., JCS 2021). The ESCRT scaffold HD-PTP has been shown to control Rab conversion via competitive scaffolding of ESCRTs and the Rabaptin-5/Rabex5 complex. Even though this mechanism and the mechanism proposed by Ott et al. are not mutually exclusive, the previous findings reduce the novelty of the present manuscript.
2. The authors found that knockdown of early ESCRT components (ESCRT-0 and ESCRT-I) prevented conversion of RAB-5 to RAB-7 on endosomes. In contrast, depletion of late ESCRT components (ESCRT-III and ESCRT-IV) resulted in co-localization of RAB-5 and RAB-7 on endosomes, and they concluded from this that early ESCRTs may act upstream of Rab conversion, while late ESCRTs might act in parallel and downstream of Rab conversion. However, previous work has shown that ubiquitinated cargos accumulate in the limiting membrane of the endosome in the absence of late as well as early ESCRT components. This contradicts the proposed model of Rabex5 recruitment due to accumulation of ubiquitinated cargos.
3. The authors have performed RNAi experiments with ESCRT-0, -I, -III and -IV. Since ESCRT-II contains ubiquitin-binding domains, the authors should also investigate the importance of this complex for Rab conversion.
4. In figure 2, the authors show that the number of ILVs is reduced and their size is increased upon *tsg-101* (RNAi). The size of MVBs in this condition is also increased. Is the overall number of MVB affected? Figure 2B would benefit from grouping the higher numbers of ILVs/MVB into one category to enlarge the figure. This would clean up the graph and still reflect the major changes happening in low ILV numbers. Do Figure 2C and figure 2A show the same microscopy? The authors provide data about an increase in MVB size upon *tsg-101* knock-down (Figure 2E-F). However, they do not comment on this. Do MVBs frequently start to cluster upon *vps-2* knockdown in the absence of SAND-1 like shown in figure 2E?
5. The authors provide information about the colocalization of different proteins by using the Mander's coefficient (e.g. Figure 3, Figure 5). Clarification for M1 and M2 in the figure would facilitate the interpretation of data.
6. The authors show an increase in colocalization between Rab5 and HGRS-1 upon knockdown of ESCRT components but no colocalization of Rab7 with HGRS-1. Did the authors test whether the change of fluorescent tag from RFP to mCherry affects the analysis? Using an RFP-Rab7 construct would clarify this. In the absence of *tsg-101*, the GFP-HGRS-1 construct does not localize to the rim of the gut lumen in Figure 3A but in 3E. Does expression of tagged Rab5 and Rab7 affect HGRS-1 localization? Did the authors test how this construct localizes without manipulation of endosomal Rab GTPases?
7. The authors express fluorescently labelled Rab GTPase constructs in *C. elegans* and HeLa cells. Does this result in high overexpression in these tissues/cells? This might make the interpretation of data concerning Rab GTPase activation and regulation more difficult. Using immunofluorescence in the mammalian system and test for endogenous Rab GTPase localization would be a nice control to strengthen the overexpression data.
8. Overall, the authors conclude strong points regarding Rab GTPase regulation from their microscopic analysis using overexpressed proteins and key endosomal proteins thereby potentially affecting the system itself. The discussion would benefit from taking this aspect into consideration. It would be very interesting to use an inducible promoter for Rab expression to exclude general effects upon long-term overexpression of these Rabs.
9. The authors conclude that VPS39 is the GEF for Rab7 in a *sand-1*(KO) condition. However, to my knowledge there is neither structural nor direct in-vitro data available which supports the idea of a direct activation of Rab7 by VPS39. With the given data this conclusion is speculative. The effect of VPS39 on Rab7 localization might be rather indirect and potentially stabilize the Rab on membranes. The authors should rephrase this part less strong. In order to conclude on the activity state of Rab7 and to distinguish from accumulation, it would be interesting to follow the localization of an effector protein of Rab7 only interacting with the active form.
10. The authors discuss that increased levels of ubiquitin on endosomes might result in stabilization of Rabex5 on membranes and impaired displacement by the Rab7 GEF. Showing this directly in their *C. elegans* system which allows to manipulate the density of ubiquitinated cargo on endosomes would strongly support this hypothesis.
11. In the mammalian system, the authors analyze the effect of a CCZ1 KO on Rab localization, for *C. elegans* the *sand-1*(KO) is analyzed. Why do the authors change their way of manipulation here? Did they also test for an effect of KO or knock down of the other subunit in one of the systems to exclude subunit-specific effects?
12. A summarizing model figure would help to integrate the new findings into context.

Minor points:

1. Microscopy images in general: Some fluorescent structures selected for the insets are very bright, are they oversaturated? This makes it hard to actually see the structures. Especially in the insets, the arrow heads and scale bars are very prominent compared to the actual image. Reducing the line widths would facilitate the view.
2. The authors chose different statistical tests to test for significance even within the same research question e.g. Figure 3C/D and F/G. A brief comment on this would help to understand.
3. The authors often combine different knockout and knockdowns of proteins for their microscopic analysis. It would be helpful to always state in the figure whether this takes place in a sand-1(KO) background and not only in the figure legend.
4. Some sentences are rather bulky, e.g. p.5: "During endosome maturation a number of processes have to happen, at least some of which should be coordinated." Please rephrase.
5. The graphs in figure 7 have a different layout compared to the other graphs, in figure 7F and 7I, the Y axis is unclearly labelled with "Area".
6. The given information for the manufacturers in the material & method section is inconsistent. Please provide specifications about the Zeiss LSM880 microscope including used objectives and the acquisition parameters of z-stacks during imaging.
7. The expression throughout the material & method section requires improvement, e.g. p. 13: "tiles showing MVBs were taken and the in this way detected MVBs were analyzed".

Referee #3:

Ott and colleagues present genetic and cell biological analyses in *C. elegans* (and in one figure, HeLa cells) exploring interactions between components of the ESCRT machinery and maturation of endosomal compartments from Rab5 to Rab7 positive. The study contains a wealth of data, is generally well-controlled (though I note as a *C. elegans* nonspecialist, I may have missed technical issues specific to the worm). Major findings include data indicating that depletion of early ESCRT proteins impairs Rab7 recruitment, while depletion of late ESCRTs allows Rab7 recruitment but impairs the loss of Rab5. Additionally, decreased total ubiquitin abundance partially relieves these phenotypes. The data are clear, suggestive, and in aggregate are a major step forward on an important problem.

Given the centrality of the Rabex-5 CUE domain in this argument, it would be useful to include an analysis of one or more ubiquitin-binding-deficient mutants.

The authors also employ genetic arguments to suggest that the HOPS complex and specifically its Vps39 subunit can act as a second GEF system for Rab7, in addition to the Ccz/Sand system. This function for Vps39 was suggested in experiments with yeast 2000 by Wurmser et al., and corroborated later by Binda et al., albeit as a claimed positive control in experiments which seemed to suggest that Vps39 is also a GEF for Rag GTPases in yeast.

Neither result has been subsequently affirmed in independent work and the most careful biochemical and genetic analyses in yeast, from Ungermann's lab, and Ungermann's and Rytka's laboratories, do not support the interpretation that Vps39 is a Rab7 GEF. Instead, Vps39 appears to be a Rab7 effector with relaxed nucleotide selectivity. Crucially, deletion of Ccz1, Mon1(SAND), or both is bypassed by Rab7(Ypt7) alleles that exhibit spontaneous nucleotide exchange activity (as first and definitively shown in forward genetic screens in Rytka's laboratory), while deletion of Vps39 and other HOPS subunits is not bypassed by these alleles. Moreover, in over a dozen papers the Wickner group's biochemical membrane fusion reconstitutions almost always contain HOPS complex known to be active. In these experiments GTP-dependent Ypt7 activity *always* requires either the presence of Ccz1-Mon1, or EDTA-mediated Mg-GNP stripping and GTP loading.

While a weak GEF activity of Vps39 is one model that could possibly explain the results in Fig. 6, the data in no way exclude alternative models. After the long confusion in this subfield, which led to almost a decade of frustrating and failed attempts to replicate the early biochemical results in multiple labs (including mine), rigorous biochemical experiments demonstrating nucleotide exchange are in my view non-negotiable. By this I mean purified proteins and quantitative assays, preferably with K_m/k_{cat} measurements and preferably corroborated by site-directed mutagenesis.

I have no problem with the experiments as shown but I do not think the interpretation is supported. I cannot sign off on the claim that Vps39 is a GEF. That claim must either be omitted or it must be rigorously substantiated with extensive and persuasive experimentation. I think the rest of the paper is easily interesting enough to support publication if the GEF claims are omitted.

Minor points: the line intensity profile plots in Figs. 5 B,F,K, 6D,H etc. are described in the legends as "quantification of area localization." They aren't. They are linear measures of intensity and appear to be representative plots rather than summaries of multiple measurements. Additionally, it's not clear in the 2D images what structures the intensity profile plots correspond to. I'm not sure what the plots add to the argument. They could be omitted, but if they are retained it would help to explain what exactly they are showing and telling us.

Referee #1:

EMBOJ-2024-117745

Mechanisms of trafficking through endosomes are intensely studied, which has given rise to a controversy between endosomal maturation versus vesicular trafficking. This has stirred investigations into molecular mechanisms as a way forward into formulating and testing precise predictions from both models. In the current manuscript, Spang and colleagues have addressed the question of how the staging between intraluminal vesicle formation and conversion from Rab5 to Rab7 endosomes could be achieved molecularly. Based on the evidence that is presented in the current manuscript, it appears that the Rab5 GEF Rabex5 and the Rab7 GEF Mon1/Ccz1/SAND1 are a key components for this staging. A model arises according to which corraling of ubiquitinated cargoes on maturing sorting endosomes liberates the ubiquitin-binding Rabex5 such that it can now be displaced by Mon1/Ccz1/SAND1 to operate Rab conversion.

This manuscript appears exceptionally mature. It addresses a timely problem from different angles, in different experimental systems and with a wealth of techniques by combining cell biological evidence from immunofluorescence and electron microscopy experiments with genetic testing. The insights from this substantial body of work into endosomal maturation are of interest to a general molecular and cellular biology readership. The manuscript is well written and pleasant to read. It provides a new model on a molecular circuit for a key step in endosomal maturation that likely operates in many organisms.

Thank you for this very positive and supportive assessment of our manuscript. We very much appreciate your comments.

Referee #2:

The manuscript of Ott et al. focuses on the spatial coordination between ESCRT function and Rab conversion during the endosomal maturation process. The authors show that loss of early ESCRT subunits leads to large Rab5-positive endosomes which do not undergo Rab conversion to Rab7 positive structures. This phenotype was alleviated by reduction of ubiquitinated cargo on endosomes. The authors further show that displacement of the Rab5 GEF RABX-5 by the Rab7 GEF SAND-1/CCZ-1 does not take place in the presence of ubiquitinated cargo. Additionally, they provide data that overexpression of Rab7 partially rescues this phenotype and suggest that the HOPS complex might as Rab7 GEF in this scenario. The authors therefore reveal a hierarchy of ESCRT recruitment and Rab conversion.

This is a thorough study that further analyzes the correlation between ILV formation and Rab conversion on endosomes to define the spatial regulation of endosomal maturation. Key data are obtained using fluorescence and transmission electron microscopy and precisely quantified. The main observations in *C. elegans* are confirmed in a mammalian cell line. Overall, this manuscript provides new insights into the special hierarchy of ILV formation and Rab conversion. However, given the methodology it is a rather descriptive study, and some of the findings reported in this manuscript are confirmatory, including the reduced ILV content and enlarged size of ESCRT-depleted endosomes, and the accumulation of ubiquitinated proteins on ESCRT-depleted endosomes. Further, the identification of VPS39 as a novel GEF directly activating Rab7 is rather speculative based on the performed microscopy experiments. The idea that Rab conversion is coordinated with ESCRT function via Rabex5 is not novel, although the proposed mechanism, Rabex5 recruitment via endosomal accumulation of ubiquitinated cargoes, is novel. However, this is a mechanism that could already be hypothesized from existing data, and the present manuscript does not provide any direct proof for it.

We thank the reviewer for the critical, but positive, assessment of our work.

Major points:

1. The issue of ESCRT function vs endosome maturation has been addressed previously (Parkinson et al., JCS 2021). The ESCRT scaffold HD-PTP has been shown to control Rab conversion via competitive scaffolding of ESCRTs and the Rabaptin-5/Rabex5 complex. Even though this mechanism and the mechanism proposed by Ott et al. are not mutually exclusive, the previous findings reduce the novelty of the present manuscript.

We respectfully disagree with the reviewer that previous findings reduce the novelty of our findings. The models are rather different and they are not mutually exclusive, which also this reviewer points out. Moreover, that a mechanism could be hypothesized based on previous findings cannot be taken as a basis that no more research should be performed on this issue. As scientists, we come up with hypotheses that we test and which either are supported by the experiments or which need to be adjusted or even turn out to be wrong; the latter being the most common outcome.

In the study mentioned by the reviewers, the authors concentrated on the role HD-PTP in endosome maturation. In their view HD-PTP is the central player, which according to their model stays engaged with ESCRT-III. We

cannot comment on this because we did not study HD-PTP in mammalian cells or its homolog EGO-2 in *C. elegans*. Likewise, Parkinson *et al.*, JCS 2021 paid a lot of attention to Rabaptin5, which we did not analyze in mammalian cells or in *C. elegans* (RABN-5). Our view is that ubiquitinated cargo availability actually is a major contributor to the timing of Rab conversion and that the lack of binding to ubiquitin destabilizes RABX-5 on membranes enabling displacement by SAND-1/CCZ-1. In Parkinson's model HD-PTP would bind ESCRT-III until ILV formation is completed. Our genetic analysis and imaging data suggest, however, that Rab conversion can take place before ILV formation is completed (see ESCRT screen and EM data). We do agree with the reviewer, however, that we could cite the Parkinson *et al.*, JCS 2021 paper, which we now do.

2. The authors found that knockdown of early ESCRT components (ESCRT-0 and ESCRT-I) prevented conversion of RAB-5 to RAB-7 on endosomes. In contrast, depletion of late ESCRT components (ESCRT-III and ESCRT-IV) resulted in co-localization of RAB-5 and RAB-7 on endosomes, and they concluded from this that early ESCRTs may act upstream of Rab conversion, while late ESCRTs might act in parallel and downstream of Rab conversion. However, previous work has shown that ubiquitinated cargos accumulate in the limiting membrane of the endosome in the absence of late as well as early ESCRT components. This contradicts the proposed model of Rabex5 recruitment due to accumulation of ubiquitinated cargos.

We respectfully disagree with the reviewer's assessment that our model contradicts findings about ubiquitinated cargo accumulation on the limiting membrane. If late ESCRTs are missing, the corraling function of the early ESCRTs is still intact and available. Thus, in our view, ubiquitinated cargoes are ushered by early ESCRTs into a subdomain of endosomes in which late ESCRTs will drive ILV formation. If late ESCRTs are lost, the formation of the subdomain of ubiquitinated cargoes is still unperturbed, yet ILVs may not form efficiently (see EM data). As suggested by this reviewer (point 12), we included a model that makes the point clearer (Fig. 9C). We did have a model in the suppl. Material, though.

We do not dispute at any point that when ILV formation is inhibited ubiquitinated cargo is still in the limiting membrane. Actually, we could also show this in our model system (Fig. 4 and Fig. EV3A-C). What we propose is that ESCRT-0 and -I bind cargo and put it into a distinct subdomain on the endosome, where normally ILV formation would occur. It is really this corraling function, which is essential and crucial to our model. In the ESCRT-0 and -I knockdowns, this corraling function is lost, cargoes are not transferred efficiently in the endosomal domain for ILV formation and hence Rabx5 will still bind to the ubiquitinated cargo and will not be displaced and therefore blocking/delaying Rab conversion. In ESCRT-III and IV knockdowns the corraling of ubiquitinated cargoes still works, and hence Rab conversion can be initiated.

To highlight the importance of cargo, we now also added data from mammalian cells, in which we show that a mutation in the ubiquitin binding domain of Rabex5 is sufficient to abolish efficient recruitment onto membranes and impairs endosomal transport (Fig. 9A and B), similar to what has been shown previously by other groups (Bonifacino, Hamacher-Brady).

3. The authors have performed RNAi experiments with ESCRT-0, -I, -III and -IV. Since ESCRT-II contains ubiquitin-binding domains, the authors should also investigate the importance of this complex for Rab conversion.

To be honest, we performed our RNAi screen also with all ESCRT components, including those of ESCRT-II. Unfortunately, we did not get any phenotypes with ESCRT-II RNAis. According to wormbase, two ESCRT-II subunits VPS-22 and VPS-36 should be essential. However, knockdowns by RNAi do not give rise to severe phenotypes and mostly mild effects on lifespan. In Roudier *et al.*, Traffic 2005, also the RNAi phenotype of ESCRT-II components was investigated. They found no phenotype for *vps-22* or *vps-25* RNAi, but reported larval lethality for a *vps-36* mutant. We invested quite a bit of time and effort to detect a phenotype after the ESCRT-II knockdowns, without any success. It was certainly not for a lack of trying. At one point, we decided to give up on ESCRT-II and proceed with the other complexes. Since we cannot absolutely rule out any technical issues from our side, we decided to not include our negative RNAi data into the manuscript.

4. In figure 2, the authors show that the number of ILVs is reduced and their size is increased upon *tsg-101* (RNAi). The size of MVBs in this condition is also increased. Is the overall number of MVB affected? Figure 2B would benefit from grouping the higher numbers of ILVs/MVB into one category to enlarge the figure. This would clean up the graph and still reflect the major changes happening in low ILV numbers. Do Figure 2C and figure 2A show the same microscopy? The authors provide data about an increase in MVB size upon *tsg-101* knock-down (Figure 2E-F). However, they do not comment on this. Do MVBs frequently start to cluster upon *vps-2* knockdown in the absence of SAND-1 like shown in figure 2E?

We did not attempt to quantify the number of MVBs. To make a conclusive statement about MVB number, we would have had to collect 3D data (either tomograms or serial sections/FIB-SEM), which we did not do. Therefore, we refrain from making any statements in this direction.

We agree with the reviewer and re-grouped the higher number ILVs together. Panel A and C do not show the same MVBs. If they appear a slightly bit larger in panel C, this is because images are slightly more enlarged. Please compare the scale bar in A and C. For the EM analysis, we used 3 worms/condition and analyzed their intestinal cells.

We do observe consistently an increase in MVB size upon *tsg-101* knockdown. Similar results have been reported in previously in other systems, which we cite in the manuscript

We do not consistently observe MVB clustering in the absence SAND-1. See also figure Fig. EV2.

5. The authors provide information about the colocalization of different proteins by using the Mander's coefficient (e.g. Figure 3, Figure 5). Clarification for M1 and M2 in the figure would facilitate the interpretation of data.

Thank you for pointing this out. We now provide the necessary information directly in the figures.

6. The authors show an increase in colocalization between Rab5 and HGRS-1 upon knockdown of ESCRT components but no colocalization of Rab7 with HGRS-1. Did the authors test whether the change of fluorescent tag from RFP to mCherry affects the analysis? Using an RFP-Rab7 construct would clarify this. In the absence of *tsg-101*, the GFP-HGRS-1 construct does not localize to the rim of the gut lumen in Figure 3A but in 3E. Does expression of tagged Rab5 and Rab7 affect HGRS-1 localization? Did the authors test how this construct localizes without manipulation of endosomal Rab GTPases?

We do not see any obvious changes in GFP-HGRS-1 localization upon expression of Rab GTPases. We have included images below, where we acquired the images side by side with exactly the same settings and they were adjusted in the same way. We apologize for the confusion caused by the differently adjusted panels. We adjusted brightness and contrast accordingly in panel E. We would like to point out that within an experiment, images are always treated the same way. This may, however, change between different experiments.

Figure 1: GFP-HGRS-1 levels do not change upon expression of RFP-RAB-5 or mCherry-RAB-7. Worms expressing either GFP-HGRS-1 alone or together with either RFP-RAB-5 or mCherry-RAB-7 were imaged side by side. We could not detect any changes in GFP-HGRS-1 protein levels.

7. The authors express fluorescently labelled Rab GTPase constructs in *C. elegans* and HeLa cells. Does this result in high overexpression in these tissues/cells? This might make the interpretation of data concerning Rab GTPase activation and regulation more difficult. Using immunofluorescence in the mammalian system and test for endogenous Rab GTPase localization would be a nice control to strengthen the overexpression data.

In *C. elegans*, we used integrated lines, but the GTPases are not endogenously tagged, and therefore the GTPases could be overexpressed. Likewise, the experiments in the mammalian cells were performed with lines stably expressing Rab GTPases, but not with CRISPR-Cas9 knock-in lines. The strains/alleles and cell lines used in our study are already well established (e.g. Solinger and Spang, Mol Biol Cell 2014, Podinovskaia *et al.*, Elife, 2021). To assess the levels of overexpression, we performed western blots from *C. elegans* lysates from wild-type and GFP-RAB-5 and mCherry-RAB-7 expressing animals. GFP-RAB-5 was about 5x overexpressed and mCherry-RAB-7 at much lower level than endogenous RAB-7 (Fig. EV4G). We conclude that we do not massively overexpress the RAB GTPases in *C. elegans*.

Similar expression levels were observed in mammalian cells stably expressing mApple-RAB5 and GFP-RAB7. The transiently transfected cells were also tested and had even lower levels of RABs than the stable cell lines (Fig. EV4H and I). Given that the stably expressing lines in *C. elegans* and in mammalian tissue culture have overexpress the RAB GTPases to a similar extent, it is tempting to speculate that their levels might be controlled

post translationally.

8. Overall, the authors conclude strong points regarding Rab GTPase regulation from their microscopic analysis using overexpressed proteins and key endosomal proteins thereby potentially affecting the system itself. The discussion would benefit from taking this aspect into consideration. It would be very interesting to use an inducible promoter for Rab expression to exclude general effects upon long-term overexpression of these Rabs.

This is an interesting idea and we will take this under consideration for future experiments. We are also currently working on knock-in lines. However, we don't have them yet and therefore cannot perform the experiments at endogenously tagged levels.

C. elegans is difficult with inducible promoters. Most commonly heat shock promoters are used, which have of course their own issues... However, considering the data provided above (point 7), it appears as if we do not massively overexpress the GTPases. We cannot exclude that the slight overexpression of GFP-RAB-5 has an effect, but it should not drastically change the outcome.

9. The authors conclude that VPS39 is the GEF for Rab7 in a *sand-1*(KO) condition. However, to my knowledge there is neither structural nor direct in-vitro data available which supports the idea of a direct activation of Rab7 by VPS39. With the given data this conclusion is speculative. The effect of VPS39 on Rab7 localization might be rather indirect and potentially stabilize the Rab on membranes. The authors should rephrase this part less strong. In order to conclude on the activity state of Rab7 and to distinguish from accumulation, it would be interesting to follow the localization of an effector protein of Rab7 only interacting with the active form.

We agree with the reviewer that we do not provide either structural or biochemical evidence for the GEF activity of VPS-39. To address this issue, we contacted Christian Ungermann and co-workers. He has purified HOPS complex, Mon1-Ccz1 and Rab7, all from yeast, though. His group performed GEF assays for us. Still, his data provide no evidence for GEF activity of the HOPS complex (personal communication). For the data, please see the response to reviewer 3. While there might still be differences between yeast and metazoan HOPS complexes, auxiliary factors, post-translational modifications, etc., we have decided to change this part of the manuscript and no longer propose that VPS-39 can act as Rab7GEF. We provide, however, ample evidence for the existence of a second Rab7GEF, of which the identity remains unknown. These data are also supported by findings in mammalian cells (Yasuda *et al.*, JCS 2016). Please see also the response to reviewer 3.

10. The authors discuss that increased levels of ubiquitin on endosomes might result in stabilization of Rabex5 on membranes and impaired displacement by the Rab7 GEF. Showing this directly in their *C. elegans* system which allows to manipulate the density of ubiquitinated cargo on endosomes would strongly support this hypothesis.

We were unable to generate a worm line expressing tagged versions of RABX-5 and SAND-1. However, Anbing Shi was kind enough to provide us with a strain in which RAB-5 is tagged with mCherry and RABX-5 with GFP. In this strain background we knocked-down either ubiquitin or HGRS-1. Reducing the ubiquitin levels, strongly impaired the endosomal localization of RABX-5 and the co-localization between RAB-5 and RABX-5. In contrast, when we knocked down HGRS-1 and thereby impaired the corralling of ubiquitinated cargo, RABX-5 strongly bound to RAB-5 positive endosomes, which were enlarged, consistent with a block in endosomes maturation. Thus, low abundance of ubiquitinated cargo will reduce the levels of RABX-5 on endosomes and high amount of ubiquitinated cargo will increase endosomal RABX-5 levels. These data are now included into the manuscript (Fig. 7F and G).

11. In the mammalian system, the authors analyze the effect of a CCZ1 KO on Rab localization, for *C. elegans* the *sand-1*(KO) is analyzed. Why do the authors change their way of manipulation here? Did they also test for an effect of KO or knock down of the other subunit in one of the systems to exclude subunit-specific effects?

We prefer KO over KDs in mammalian cells because the phenotype is cleaner. We changed from Mon1 to Ccz1 because there is MON1a and b in humans which can partially compensate for each other (Poteryaev *et al.*, Cell 2010, Hiragi *et al.*, JCS 2022). Moreover, it also appears that there are more copies of MON1 due to aberrant chromosome numbers in HeLa. This does not seem to be the case for CCZ1. This is why we changed the target. We have already used this cell line in a previous publication to investigate the connection between Rab conversion and endosomal acidification (Podinovskaia *et al.*, Elife 2021). In *C. elegans*, the phenotypes for *sand-1* and *ccz-1* mutants appear to be the same and the combining both mutations did not increase the severity of the phenotype. Similarly, $\Delta mon1$ and $\Delta ccz1$ share the same phenotypes (C. Ungermann pers. communication). We have, therefore, no reason to believe that the KO of CCZ1 would not reflect the loss of MON1/CCZ1 Rab7GEF activity.

12. A summarizing model figure would help to integrate the new findings into context.

We had a summarizing model figure in the supplemental material. We simplified the model and concentrated on the role of ubiquitinated cargo and its segregation into a subdomain to drive Rab conversion (Fig. 9C).

Minor points:

1. Microscopy images in general: Some fluorescent structures selected for the insets are very bright, are they oversaturated? This makes it hard to actually see the structures. Especially in the insets, the arrow heads and scale bars are very prominent compared to the actual image. Reducing the line widths would facilitate the view.

We took care that there is no oversaturation of the images and reduced the line width and the size of the arrowheads and scale bars.

2. The authors chose different statistical tests to test for significance even within the same research question e.g. Figure 3C/D and F/G. A brief comment on this would help to understand.

We provide now more information on the choice of the statistical test in the Materials and Methods section.

3. The authors often combine different knockout and knockdowns of proteins for their microscopic analysis. It would be helpful to always state in the figure whether this takes place in a *sand-1(KO)* background and not only in the figure legend.

We mention now in the figure and not only in the legend when the *sand-1(KO)* strain was used.

4. Some sentences are rather bulky, e.g. p.5: "During endosome maturation a number of processes have to happen, at least some of which should be coordinated." Please rephrase.

We aimed to reduce the bulkiness of some sentence and use simple language.

5. The graphs in figure 7 have a different layout compared to the other graphs, in figure 7F and 7I, the Y axis is unclearly labelled with "Area".

We harmonized the layouts of the figures.

6. The given information for the manufacturers in the material & method section is inconsistent. Please provide specifications about the Zeiss LSM880 microscope including used objectives and the acquisition parameters of z-stacks during imaging.

We added the requested information to the Materials and Methods section.

7. The expression throughout the material & method section requires improvement, e.g. p. 13: "tiles showing MVBs were taken and the in this way detected MVBs were analyzed".

We improved the language of the Materials and Methods part.

Referee #3:

Ott and colleagues present genetic and cell biological analyses in *C. elegans* (and in one figure, HeLa cells) exploring interactions between components of the ESCRT machinery and maturation of endosomal compartments from Rab5 to Rab7 positive. The study contains a wealth of data, is generally well-controlled (though I note as a *C. elegans* nonspecialist, I may have missed technical issues specific to the worm). Major findings include data indicating that depletion of early ESCRT proteins impairs Rab7 recruitment, while depletion of late ESCRTs allows Rab7 recruitment but impairs the loss of Rab5. Additionally, decreased total ubiquitin abundance partially relieves these phenotypes. The data are clear, suggestive, and in aggregate are a major step forward on an important problem.

Thank you, Alex for the positive assessment of our work!

Given the centrality of the Rabex-5 CUE domain in this argument, it would be useful to include an analysis of one or more ubiquitin-binding-deficient mutants.

Thank you for the suggestion. We obtained RABEX5 ubiquitin binding mutants from Juan Bonifacio and determined the localization of them in HeLa cells. As reported previously, impaired ubiquitin binding results in an increase of cytoplasmic RABEX5, with very little on endosomes. While overexpression of WT Rabex-5 caused the enlargement of early endosomes and more RABEX5 binding. These data are now incorporated into the manuscript (Fig. 9A and B) and complementary to new data, which we now provide in *C. elegans* (Fig. 7F and G). There, we assessed RABX-5 localization under low or high abundance of ubiquitinated cargoes on endosomes by either downregulating ubiquitin or HGRS-1. Low abundance of ubiquitinated cargo resulted in cytoplasmic RABX-5 staining, while high abundance of ubiquitinated cargo caused massive recruitment of RABX-5 onto early endosomes and their enlargement.

The authors also employ genetic arguments to suggest that the HOPS complex and specifically its Vps39 substituent can act as a second GEF system for Rab7, in addition to the Ccz/Sand system. This function for Vps39 was suggested in experiments with yeast 2000 by Wurmser et al., and corroborated later by Binda et al., albeit as a claimed positive control in experiments which seemed to suggest that Vps39 is also a GEF for Rag GTPases in yeast.

Neither result has been subsequently affirmed in independent work and the most careful biochemical and genetic analyses in yeast, from Ungermann's lab, and Ungermann's and Rytka's laboratories, do not support the interpretation that Vps37 is a Rab7 GEF. Instead, Vps39 appears to be a Rab7 effector with relaxed nucleotide selectivity. Crucially, deletion of Ccz1, Mon1(SAND), or both is bypassed by Rab7(Ypt7) alleles that exhibit spontaneous nucleotide exchange activity (as first and definitively shown in forward genetic screens in Rytka's laboratory), while deletion of Vps39 and other HOPS subunits is not bypassed by these alleles. Moreover, in over a dozen papers the Wickner group's biochemical membrane fusion reconstitutions almost always contain HOPS complex known to be active. In these experiments GTP-dependent Ypt7 activity *always* requires either the presence of Ccz1-Mon1, or EDTA-mediated Mg-GNP stripping and GTP loading.

While a weak GEF activity of Vps39 is one model that could possibly explain the results in Fig. 6, the data in no way exclude alternative models. After the long confusion in this subfield, which led to almost a decade of frustrating and failed attempts to replicate the early biochemical results in multiple labs (including mine), rigorous biochemical experiments demonstrating nucleotide exchange are in my view non-negotiable. By this I mean purified proteins and quantitative assays, preferably with Km/kcat measurements and preferably corroborated by site-directed mutagenesis.

I have no problem with the experiments as shown but I do not think the interpretation is supported. I cannot sign off on the claim that Vps39 is a GEF. That claim must either be omitted or it must be rigorously substantiated with extensive and persuasive experimentation. I think the rest of the paper is easily interesting enough to support publication if the GEF claims are omitted.

There are a number of reasons why there needs to be a second Rab7GEF at least in *C. elegans* and probably also in mammals and why we thought it could be VPS-39

- 1) RAB-7 is essential in *C. elegans*, while SAND-1/CMON-1 and CCZ-1 are not. The animals are very sick and temperature-sensitive embryonic lethal (this is how we cloned *sand-1* initially) (Poteryaev and Spang, Biochem Soc Trans 2005).
- 2) When we imaged Rab conversion in *C. elegans*, we also imaged the recruitment of SAND-1 onto endosomes. We found that SAND-1 and RAB-7 were recruited roughly at the same time, SAND-1 often a bit earlier, but we could not really resolve this very well back then. Importantly, SAND-1 left the endosomes after about 1-2 min, while RAB-7 levels continued to raise afterwards and RAB-7 stayed on for a long time (Poteryaev et al., Cell 2010).
- 3) The Ungermann lab (Nordmann et al., Curr Biol 2010) examined the Rab7GEF activity by treating vacuoles with a Rab7 GAP to remove all Ypt7 from vacuoles and then added increasing concentrations of either HOPS or Mon1-Ccz1 together with Ypt7 to these vacuoles and tested Rab recruitment. In this setup, they showed that at low concentrations HOPS was as sufficient as Mon1-Ccz1 in recruiting Ypt7 onto vacuoles. Only at high concentrations HOPS was unable to recruit Ypt7.
- 4) Neither *sand-1(KO)*, *ccz-1(KO)* nor the double KO are lethal. They all have the same phenotype: temperature sensitive embryonic lethality. As stated above RAB-7 is essential in *C. elegans*.
- 5) And of course, the data from this manuscript: in a *sand-1(KO)*, RAB-7 recruitment onto endosomes and endosome maturation can be partially rescued.
- 6) Yasuda et al., JCS 2016 reported that Mon1/Ccz1 was unable to recruit Rab7 onto lysosomes.

I discussed our data with Christian Ungermann and his postdoc Lars Langemeyer. They agreed to revisit the issue and performed GEF assays with MANT-GDP. Their results show no Rab7GEF activity for the yeast HOPS complex. Figure below (data by Langemeyer and Ungermann; personal communication).

There is of course the caveat that this could be different in metazoans or that there are post-translational modifications/an auxiliary protein that would change activity. At this point this is too much speculation. Therefore, we no longer claim that VPS-39 is a second Rab7GEF and re-wrote this part. However, we hope you agree that we provide striking evidence for the presence of a second Rab7GEF.

Figure 2: MANT-GDP assay to measure GEF activity of Mon1/Ccz1 and HOPS. Liposomes were loaded with either prenylated Ypt10-GTP (for Mon1/Ccz1) or prenylated Ypt7-GTP (for HOPS) and then the GEFs together with Ypt7-(MANT)GDP were added. While loss of MANT-GDP fluorescence -indicative of GEF activity- was observed in the presence of Mon1/Ccz1, no drop in fluorescence was observed in the presence of HOPS, indicating that HOPS by itself is not able to act as a Rab7GEF.

Minor points: the line intensity profile plots in Figs. 5 B,F,K, 6D,H etc. are described in the legends as "quantification of area localization." They aren't. They are linear measures of intensity and appear to be representative plots rather than summaries of multiple measurements. Additionally, it's not clear in the 2D images what structures the intensity profile plots correspond to. I'm not sure what the plots add to the argument. They could be omitted, but if they are retained it would help to explain what exactly they are showing and telling us.

We fixed the description of the measurements and added a small diagram to explain what was measured to Fig. 5.

-Alex Merz

Dear Anne,

Thank you for submitting the revised version of your manuscript, which addresses the concerns of the referees. This revised version has now been re-reviewed; I attach the second referee reports to the bottom of this mail. As you will see, you have addressed the referees' concerns to their satisfaction. Reviewer 2 makes some final constructive suggestions which I would like you to consider carefully. Before I can finally accept the manuscript, there are some remaining editorial points which need to be addressed. In this regard, would you please:

- limit the number of keywords to seven,
- include a Data Availability section that contains the following statement: 'This study includes no data deposited in external repositories',
- complete the 3rd (pink) column in the author checklist for all positive responses; they should be left blank if the response is "Not Applicable",
- use the nomenclature Figure EV# in figure legends in the manuscript file,
- upload Appendix file in PDF format; nomenclature should be Appendix Table S# (not Table S#) throughout Appendix PDF and manuscript file; the first page of the Appendix file should have a table of contents with page numbers,
- provide source data for Fig. 1A, 1B, 2A, 2C, 2E, 3A, 3E, 4A, 5A, 5D, 5I, 6A, 6C, 7B, 8A, 8D, 8E, 8H; save files in a scheme of one figure/folder and then uploaded as .zip files. E.g. all the Source data files for figure 1 need to be saved in a single folder and this needs to be zipped and then uploaded as "SD figure 1.zip" file. For EV and/or appendix figures, ZIP together all source data,
- add a 'Reagents and Tools' table,
- provide exact p values are not provided in the legends of figures 2d, f; 3c-d, g; 4b; 5g; 6b, e; 7e, g; 8b, f, i; 9b; EV 3c,
- correct the mismatch between the annotated p values in the figure legend and the annotated p values in the figure file in figures 3c-d, g; 4b,
- define n in the legends of figures 6b, e, and
- define the white/yellow arrowheads in the legends of figure 1a, 9a, EV 3b

I look forward to receiving these changes. EMBO Press is an editorially independent publishing platform for the development of EMBO scientific publications.

Best wishes,

William

William Teale, PhD
Editor
The EMBO Journal
w.teale@embojournal.org

We realize that it is difficult to revise to a specific deadline. In the interest of protecting the conceptual advance provided by the work, we recommend a revision within 3 months (2nd Mar 2025). Please discuss the revision progress ahead of this time with the editor if you require more time to complete the revisions. Use the link below to submit your revision:

Link Unavailable

Referee #1:

I had already recommended publication of the initial version of the manuscript.

Referee #3:

The authors have addressed my concerns, particularly about the putative Rab7 GEF activity of Vps39. I would prefer that some of the remaining speculation about how the results might best be interpreted be confined to the Discussion rather than being presented in the Results. I leave that decision to the authors and monitoring Editor.

I did notice on this reading some omissions from the cited literature, specifically work from several labs showing that in flies and mammals, HOPS seems to be recruited by Arl8 as well as by Rab7. So another model might be that Arl8-mediated HOPS recruitment captures rare spontaneously exchanging Rab7:GTP and blocks GAP-mediated hydrolysis. The phenotypes observed for Rab7 overexpression in the present study might be consistent with such an alternative interpretation. It would be worth citing one or more of the relevant papers.

PMID: 36640308

PMID: 30115618

PMID: 24501423

PMID: 25908847

Nevertheless, as stated above my concerns are generally addressed.

-Alex Merz

limit the number of keywords to seven,

In the manuscript, we only used seven keywords. We will reduce the number of keywords online during resubmission.

- include a Data Availability section that contains the following statement: 'This study includes no data deposited in external repositories',

Actually, the last version of the manuscript contained already this statement. Now that you ask us to deposit our data into a repository, we have amended the statement to: This data includes data deposited into BioImage under accession number S-BIAD1503.

- complete the 3rd (pink) column in the author checklist for all positive responses; they should be left blank if the response is "Not Applicable",

We amended the author checklist.

- use the nomenclature Figure EV# in figure legends in the manuscript file,

We corrected the calling of Figures in the figure legends

- upload Appendix file in PDF format; nomenclature should be Appendix Table S# (not Table S#) throughout Appendix PDF and manuscript file; the first page of the Appendix file should have a table of contents with page numbers,

We prepared the Appendix file in pdf format, used the correct nomenclature and provide a table of contents with page numbers.

- provide source data for Fig. 1A, 1B, 2A, 2C, 2E, 3A, 3E, 4A, 5A, 5D, 5I, 6A, 6C, 7B, 8A, 8D, 8E, 8H; save files in a scheme of one figure/folder and then uploaded as .zip files. E.g. all the Source data files for figure 1 need to be saved in a single folder and this needs to be zipped and then uploaded as "SD figure 1.zip" file. For EV and/or appendix figures, ZIP together all source data,

We provided already the source data in the last submission. We had to reduce the quality of the EM part because of the image size. The EM data have been uploaded on the BioImage server and the accession number is provided in the manuscript. We also followed the instructions for the source data.

- add a 'Reagents and Tools' table,

We now add a Reagents and Tools table

- provide exact p values are not provided in the legends of figures 2d, f; 3c-d, g; 4b; 5g; 6b, e; 7e, g; 8b, f, i; 9b; EV 3c,

We provide now the exact p values.

- correct the mismatch between the annotated p values in the figure legend and the annotated p values in the figure file in figures 3c-d, g; 4b,

We apologize for the mistake, which has been corrected.

- define n in the legends of figures 6b, e, and

We now provide the n for both panels.

- define the white/yellow arrowheads in the legends of figure 1a, 9a, EV 3b

We apologize for the omission. We now define the white and yellow arrowheads.

Referee #3:

We thank Alex Merz for the positive assessment.

The authors have addressed my concerns, particularly about the putative Rab7 GEF activity of Vps39. I would prefer that some of the remaining speculation about how the results might best be interpreted be confined to the Discussion rather than being presented in the Results. I leave that decision to the authors and monitoring Editor.

We considered changing the wording in the result part. However, this would have disturbed the flow of the manuscript and therefore we left this part as is.

I did notice on this reading some omissions from the cited literature, specifically work from several labs showing that in flies and mammals, HOPS seems to be recruited by Arl8 as well as by Rab7. So another model might be that Arl8-mediated HOPS recruitment captures rare spontaneously exchanging Rab7:GTP and blocks GAP-mediated hydrolysis. The phenotypes observed for Rab7 overexpression in the present study might be consistent with such an alternative interpretation. It would be worth citing one or more of the relevant papers.

PMID: 36640308

PMID: 30115618

PMID: 24501423

PMID: 25908847

We do not think that a discussion of the Arl8/HOPS papers is actually necessary. The likelihood that HOPS can capture a spontaneously activated Rab7 is extremely low. The intrinsic GDP to GTP exchange activity of Rab7 is negligible. Even if this there would be a Rab7:GTP magically

appear close to HOPS and be captured, one would still need a positive feedback loop to recruit sufficient amounts of Rab7 to fulfill its function on late endosomes, endolysosomes and lysosomes. Nevertheless, to appease the reviewer, we now mention this possibility in the discussion and cite the relevant papers.

Nevertheless, as stated above my concerns are generally addressed.

Thank you!

-Alex Merz

Dear Anne,

I am pleased to inform you that your manuscript has been accepted for publication in the EMBO Journal.

Congratulations!

Yours sincerely,

William

William Teale, PhD
Editor
The EMBO Journal
w.teale@embojournal.org
